



# Rapid transformation of ambient absorbing aerosols from West African biomass burning

Huihui Wu[1], Jonathan W Taylor[1], Justin M Langridge[2], Chenjie Yu[1], James D Allan[1, 3], Kate Szpek[2], Michael I Cotterell[4, *], Paul I Williams[1, 3], Michael Flynn[1], Patrick Barker[1], Cathryn Fox[2], Grant Allen[1], James Lee[5,6], and Hugh Coe[1]

[1]Department of Earth and Environmental Sciences, University of Manchester, Manchester, UK
[2]Met Office, Fitzroy Road, Exeter, EX1 3PB, UK
[3]National Centre for Atmospheric Science, University of Manchester, Manchester, UK
[4]College of Engineering, Mathematics and Physical Sciences, University of Exeter, Exeter, UK.
[5]Wolfson Atmospheric Chemistry Laboratories, Department of Chemistry, University of York, York YO10 5DD, UK
[6]National Centre for Atmospheric Sciences, University of York, York YO10 5DD, UK
[*]Now at School of Chemistry, University of Bristol, Bristol, United Kingdom

*Correspondence to*: Hugh Coe (hugh.coe@manchester.ac.uk)

**Abstract.** Seasonal biomass burning (BB) over West Africa is a globally significant source of carbonaceous particles in the atmosphere, which have important climate impacts but are poorly constrained. Here, the evolution of smoke aerosols emitted from flaming-controlled burning of agricultural waste and wooded savannah in the Senegal region was characterized over a timescale of half-day advection from source during the MOYA-2017 (Methane Observation Yearly Assessment-2017) aircraft campaign. Plumes from such fire types are rich in black carbon (BC) emissions. Concurrent measurements of chemical composition, organic aerosol (OA) oxidation state, bulk aerosol size and BC mixing state reveal that emitted BB submicron aerosols changed dramatically with time. Various aerosol optical properties (e.g., absorption Ångström exponent (AAE), and mass absorption coefficients (MAC)) also evolved with ageing. In this study, brown carbon (BrC) was a minor fractional component of the freshly emitted BB aerosols (< 0.5 h), but the increasing AAE with particle age indicates that BrC formation dominated over any loss process over the first ~12 hours of plume transport. Using different methods, the fractional contribution of BrC to total aerosol absorption showed an increasing trend with time and was ~ 18–31 % at the optical wavelength of 405 nm after half-day transport. The generated BrC was found to be positively correlated with oxygenated and low-volatility OA, likely from the oxidation of evaporated primary OA and secondary OA formation. We found that the evolution of BrC with particle age was different in this region compared with previous BB field studies that mainly focused on emissions from smouldering fires, which have shown a high contribution from BrC at source and BrC net loss upon ageing. This study suggests an initial stage of BrC absorption enhancement during the evolution of BB smoke. Secondary processing is the dominant contributor to BrC production in this BB region, in contrast to the primary emission of BrC previously reported in other BB studies. The total aerosol absorption normalized to BC mass ($MAC_{meas-BC}$) was also enhanced with ageing, due to the lensing effect of increasingly thick coatings on BC and the absorption by BrC. The effect of ageing on aerosol absorption, represented by the absorption enhancement ($E_{Abs-MAC}$), was estimated over timescales of



hours. MOYA-2017 provides novel field results. The comparisons between MOYA-2017 and previous field studies imply

that the evolution of absorbing aerosols (BC and BrC) after emission vary with source combustion conditions. Different treatments of absorbing aerosol properties and their evolution in different types of fires should be considered when modelling regional radiative forcing. These observational results will be very important for predicting climate effects of BB aerosol in regions controlled by flaming burning of agricultural waste and savannah-like biomass fuels.

## 1. Introduction

Biomass burning (BB) of agricultural waste and savannah in the sub-Sahelian regions of West Africa during the dry season (November to February) is a strong contributor to the global aerosol burden every year (Roberts et al., 2009; Andreae, 2019). Emitted BB plumes over West Africa generally move southwest across the continent and are then transported over the North Atlantic Ocean. Sometimes, weak southerly advection over the land surface can drive air to move northward and the warm BB plumes tend to be lifted over the cooler and drier Saharan air. Under this latter scenario, the upper-level circulation

plays a role in transporting these lifted plumes southward toward the Atlantic Ocean (Haywood et al., 2008). These BB aerosols have an important impact on the regional climate by scattering and absorbing solar radiation, and also interact with clouds. The overall climate effects of these BB aerosols are a combination of interacting warming and cooling effects, depending on the aerosol vertical distribution and their relative locations with respect to clouds, as well as their chemical, physical and optical properties and evolution with transport time (e.g. Boucher et al., 2013).

Field measurements of African BB indicate that BB aerosols are dominated by carbonaceous particles including those comprised of black carbon (BC) and organic aerosol (OA) components, with lesser contributions from inorganic species (Capes et al., 2008, Vakkari et al., 2014). BC is thought to be the main absorbing aerosol in BB smoke, directly absorbing radiation across the solar spectrum (Bond et al., 2013). Particles composed of only OA predominantly have a cooling effect by efficiently scattering radiation in the solar spectrum. However, certain types of OA, known as "brown carbon" (BrC) also

absorb solar radiation in the near-ultraviolet (near-UV, 300−400 nm) and visible (400−700 nm) ranges, although this absorption is strongly wavelength dependent compared to the absorption spectrum for BC (Laskin et al., 2015). Moreover, OA can contribute to enhanced aerosol absorption when OA is internally mixed with BC, through the so-called "lensing effect" (Liu et al., 2017). The inclusion of BrC in global climate simulations is not common; one study has suggested an average global direct radiative forcing (DRF) of BrC as +0.13 W $m^{-2}$ (Brown et al., 2018), which is ~ 20% of the global

DRF of BC (+0.71 W $m^{-2}$) estimated by Bond et al. (2013), though these estimates are both associated with considerable uncertainty. Regional effects of BrC over major areas of BB such as subtropical Africa, may be substantially larger than this (Feng et al., 2013), necessitating consideration of both BC and BrC absorption in the West Africa region.

Previous studies have characterised freshly emitted BB aerosols to some extent. The initial relative contribution of OA and BC varies widely with fuel type and combustion conditions, as does the corresponding initial aerosol size distribution

(Vakkari et al., 2014). The initial optical properties (i.e. single scattering albedo (SSA), absorption Ångström exponent





(AAE) and mass absorption coefficient (MAC) of BC and BrC) and BC mixing states of freshly emitted BB aerosols depend strongly on OA/BC mass ratios and combustion efficiency (Liu S. et al., 2014; Saleh et al., 2014; Pokhrel et al., 2016, 2017; McClure et al., 2020) and are therefore also highly variable. The properties of BB aerosols have been shown to evolve over time post-emission. Understanding this ageing process is vital to evaluate their atmospheric impacts. The chemical and size

evolution of BB aerosols have been studied comprehensively in field and laboratory measurements under various BB conditions (e.g. Capes et al., 2008; Yokelson et al., 2009; Cubison et al., 2011; Pratt et al., 2011; Akagi et al., 2012; Ortega et al., 2013; Vakkari et al., 2018; Kleinman et al., 2020). However, studies of optical evolution are limited, especially in field observations. Existing measurements show that the absorbing properties of BC are modified after emission, due to the internal mixing of BC with other species such as inorganics and OA (Bond et al., 2013). The MAC of BC may be enhanced

by a lensing effect induced by the coatings and/or the absorption from BrC (Healy et al., 2015). The absorbing properties of BrC are also modified with ageing, closely related to secondary BrC formation from photochemical processing of co-emitted gaseous compounds (Saleh et al., 2013; Palm et al., 2020) and loss by photobleaching (Lee et al., 2014; Zhao et al., 2015). The normalized excess aerosol scattering with respect to a BB tracer (carbon monoxide, CO) is shown to increase during atmospheric ageing due to aerosol growth caused by condensation and coagulation, resulting in an increasing SSA

downwind from source (Abel et al., 2003; Yokelson et al., 2009; Akagi et al., 2012; Vakkari et al., 2014; Kleinman et al., 2020). The accurate characterization and application of these optical properties and their evolution with time under ambient conditions are key issues in modelling the climate effects of BB aerosols.

Laboratory experiments have been used to simulate the photochemical ageing of solution-extracted particles or smoke (particles plus gases) generated from burning various biomass fuel types under different combustion conditions. These

experiments have indicated that the evolution of BC mixing state and optical properties (e.g. SSA, AAE, and MAC) for BB aerosols vary with initial emission conditions (Zhong and Jang, 2014; Wong et al., 2017, 2019; Kumar et al., 2018; Cappa et al., 2020). These results provide an insight into the behaviour of BB aerosols in the ambient atmosphere; region-specific characterizations of the evolution of aerosol optical properties are crucial for improving descriptions in atmospheric models, owing to the diversity in the ageing process of BB aerosols between wildfire sources. Current field observations

demonstrating this evolution are sparse, particularly the effects of ageing on the light absorption properties of BrC and BC. Some field studies have reported enhanced absorption of internally mixed BC in aged BB smoke far from source (e.g. Lack et al., 2012a), but the evolution trends of BC mixing state and optics are quantified poorly. Forrister et al. (2015) tracked wildfire plumes over North America and reported decreasing AAE and the loss of BrC over ~2 days of atmospheric transport. Wang et al. (2016) observed a decreasing MAC for BrC with a lifetime of ~1 day during the Amazonian BB

season near Manaus, Brazil. These field measurements covered only a limited range of combustion fuels and conditions. Additional field observations of the evolution of BB absorptivity are necessary to extend laboratory mechanisms, to understand ambient atmospheric processes, and to provide observational constraints for atmospheric models.

Although West Africa is one of the most important BB regions on a global scale, field observations of BB aerosols in this region are limited. The Dust and Biomass Experiment (DABEX) in January to February 2006 took place in West Africa





and investigated the chemical composition, size distribution and optics of BB aerosols at different plume stages (close to source, elevated BB layer and transported BB layer) (Capes et al., 2008; Johnson et al., 2008). During the DABEX campaign, aerosol absorption and scattering were measured using a filter-based technique (Particle Soot Absorption Photometer) and a Nephelometer, respectively. Limitations with filter-based measurements of aerosol light absorption were reported in previous studies (Lack et al., 2008; Davies et al., 2019), and aerosol absorption was characterized at only a single

wavelength of 567 nm during the DABEX. BC concentrations were estimated using an indirect method by multiplying the measured absorption (at a single wavelength) with an assumed mass absorption coefficient of 12 $m^2$ $g^{-1}$ for soot. The properties of BB aerosols, especially the optical properties and BC microphysical properties were characterized with high uncertainties during the DABEX, owing to the lack of advanced measurement systems that have only become available in the last decade. To provide accurate BB aerosol parameterizations for modelling regional radiation interactions over the

important West Africa region, we report new field measurements acquired during the MOYA-2017 (Methane Observation Yearly Assessment-2017) aircraft campaign, benefitting from significant advances in measurement techniques. Here, we present the evolution of BB aerosols over the first ~12 h post-emission, including a detailed characterization of chemical, microphysical, and optical properties. In addition, we combined measurements with optical modelling to investigate the effects of ageing on the light absorption properties of BrC and BC in this region. We also investigated the relationship

between chemical composition and BrC optical properties.

## 2. Experimental

The research flights during MOYA-2017 (Methane Observation Yearly Assessment-2017) were made by the UK Facility for Airborne Atmospheric Measurements (FAAM), using the BAe-146 Atmospheric Research Aircraft (ARA). The flights sampled freshly emitted plumes from wildfires over the Senegal area, in addition to aged smoke transported

southwest over the continent and the Atlantic Ocean. Nearby background air out of the plume was also sampled. The aircraft included a range of in situ instruments to measure aerosol composition, size distribution and optical properties, trace gas concentrations and standard meteorological variables. A further description of the MOYA-2017 campaign is reported in Barker et al., 2020. Tracks of the flights (with flight numbers labelled from C005 to C007) used in this study are shown in Fig.1a. Detailed information about the selected smoke plumes is provided in Sect. 3.1. The main instruments used in this

study are described below.

### 2.1 Airborne measurements

Refractory black carbon (BC) concentrations and physical properties were measured using a single particle soot photometer (SP2), the operation of which on the ARA has been described by McMeeking et al. (2010). The SP2 detects BC-containing particles with an equivalent spherical diameter in the range of 70 – 850 nm (Liu, D. et al., 2014). Briefly, an intra-

cavity Nd-YAG laser beam at λ = 1064 nm is used for particle detection on a single particle basis. The laser beam heats





particles containing BC to their incandescence temperature and visible light is emitted. The SP2 incandescence signal is used to derive the mass of refractory BC present in the particle, and the mass can be converted to a spherical-equivalent BC core diameter ($D_C$) with an assumed BC density of 1.8 g cm$^{-3}$ (Bond et al., 2013). Aquadag BC particle standards were used to calibrate the SP2 incandescence signal during the campaign, following the calibration procedures in Laborde et al (2012).

The SP2 can also provide information on the coating properties for BC-containing particles, by determining the $\lambda$ = 1064 nm light scattering cross-section of particles. The scattering signal of a BC-containing particle will be distorted during its transit through the laser beam caused by the mass loss of a BC-containing particle from laser heating. The methodology of leading edge only (LEO) fitting was used to reconstruct the scattering signal of a BC-containing particle, as described in detail in previous studies (Gao et al., 2007; Liu, D. et al., 2014; Taylor et al., 2015). The scattering channel was calibrated using

polystyrene latex spheres at sizes of 200 and 300 nm. A Mie core/shell model was then used to infer the coated particle diameter ($D_P$) and hence the shell/core ratios ($D_P/D_C$), which depends on the assumed refractive index of the BC core and coating (Taylor et al., 2015). This model and the required assumptions on refractive indices are further discussed in Sect. 2.3. The SP2 single-particle data were also examined for coincidence at high concentrations, following the procedures described in Taylor et al. (2020).

A compact time-of-flight aerosol mass spectrometer (C-ToF AMS) was used to measure the mass concentrations of non-refractory aerosols, including OA, sulfate, nitrate, ammonium and a fraction of chloride (Drewnick et al., 2005). The AMS collects sub-micron particles in the aerodynamic diameter range of ~50–800 nm via an aerodynamic lens. The instrument setup and calibration procedure on the ARA have been described by Morgan et al. (2009). The AMS was calibrated using mobility-size-selected ammonium nitrate and ammonium sulfate particles. The AMS data was processed

using the standard SQUIRREL (SeQUential Igor data RetRiEvaL, v.1.60N) ToF-AMS software package. A time and composition dependent collection efficiency (CE) was applied to the data based on the algorithm by Middlebrook et al. (2012). The 2σ uncertainties in AMS-measured mass concentrations during aircraft operation were estimated as 30% by Bahreini et al. (2009). The AMS data for flight C006 is not available as the vacuum pump overheated during this flight. In this study, the OA fragment markers (m/z 43, 44, 57 and 60) and proportional contributions of OA fragment markers to the

total OA mass (*f*43, *f*44, *f*57 and *f*60) were calculated. Oxygen/carbon (O:C) ratio and the ratio of organic mass to organic carbon (OM/OC) were estimated following the methods developed by Aiken et al. (2008). OM is the total mass of the OA and OC is the mass of carbon associated with the OA.

The dry (<10 % RH) aerosol absorption coefficients (B$_{Abs}$, Mm$^{-1}$) were measured using a custom-built suite of multi-wavelength photoacoustic spectrometers (PAS, at wavelengths $\lambda$ = 405, 514, 658 nm), providing direct and non-contact

measurements of aerosol absorption. The setup and calibration of these instruments are described in detail by Davies et al. (2018, 2019) and Cotterell et al. (2020). An impactor removed particles with aerodynamic diameters > 1.3 μm. The accuracy of the PAS calibration was better than 8% (Davies et al., 2018). The absorption Ångström exponent was determined by the equation:



$$AAE = -\left|\frac{\ln(B_{Abs}(\lambda_2)) - \ln(B_{Abs}(\lambda_1))}{\ln(\lambda_2) - \ln(\lambda_1)}\right| \tag{1}$$

in which $\lambda$ is the wavelength. The uncertainty in the determined AAE is expected to be < 5% (Taylor et al., 2020).

Aerosol number size distribution was measured using an on-board Scanning Mobility Particle Sizer (SMPS). The SMPS sampled aerosols from the same inlet as the AMS, and measured distributions of particle mobility diameter in the range of 20–350 nm. A low-pressure water-based condensation particle counter (WCPC model 3786-LP) was connected to a TSI 3081 differential mobility analyser (DMA). Following the schemes developed by Zhou (2001), the SMPS data were inverted based on a ~1 min averaging time when aerosol mass concentrations from the AMS and SP2 varied less than 30%. CO was measured using an AeroLaser AL5002 Vacuum-UV fast fluorescence instrument, with an accuracy of ±3% and precision of 1 ppb (Gerbig et al., 1999). Carbon dioxide ($CO_2$) and methane ($CH_4$) were measured using a Fast Greenhouse Gas Analyser (FGGA) (O'Shea et al., 2013) and was calibrated using gas standards traceable to the WMO-X2007 scale (Barker et al., 2020).

All measurements reported for aerosols and gases were corrected to standard temperature and pressure (STP, 273.15K and 1013.25 hPa). The SP2, PAS, CO, and FGGA data were recorded at a 1-Hz sampling frequency, while the AMS sampling frequency was either 8 or 15 s. Related calculation methods are listed in detail in Sect. 3 and Sect. S2 of Supplementary Information.

**2.2 Back-trajectory calculations**

We used the UK Met Office's Numerical Atmospheric Modelling Environment (NAME) (Jones et al., 2007) to track the history of sampled air masses over the Atlantic Ocean. The inert particles were released along the aircraft track every 30s, and their 1-day back trajectories were modelled using three-dimensional gridded meteorological fields derived from the UK Met Office's global Numerical Weather Prediction model, the Unified Model (Brown et al., 2012). These fields are updated every 3 h and have a high resolution of 0.23° longitude by 0.16° latitude. The meteorological fields have 59 vertical levels up to an approximate height of 29 km. The NAME model was chosen for this study because it uses high-resolution meteorological data of approximately 17 km × 17 km, and it can predict dispersion over distances ranging from a few kilometres to the whole globe. The fire sources were identified based on the Collection 6 Terra and Aqua Moderate Resolution Imaging Spectroradiometer (MODIS) fire products (Giglio et al., 2018). Air mass transport times, in hours since emission, were estimated from the point of aircraft measurements to the possible fire sources.

**2.3 Optical modelling**

In this study, we simulated the MAC and AAE of coated BC with non-absorbing coatings, using a variety of optical models. Firstly, we determined the size and mixing state of BC-containing particles from the single-particle measurements of BC core mass ($M_{BC}$) and scattering cross-section from the SP2. This process is based on previous works of Taylor et al. (2015) and Liu et al., (2017). Taylor et al. (2015) described the steps to calculate the single-particle spherical-equivalent $D_C$





and $D_P$, with the SP2 measurements and a scattering model using core/shell Mie theory. An empirical correction recommended by Liu et al. (2017) was added to the data processing steps described in Taylor et al. (2015). We converted the calculated single-particle $D_P/D_C$ ratio to the mass ratio of non-BC to BC (MR) and generated a 2-D distribution of MR vs. $M_{BC}$. Further details on the processing of SP2 data are provided in Sect. S2.1 of the Supplementary Information. For SP2 measurements, not all detected particles have a successful LEO fitting to measure the scattering cross-section of BC-

containing particles at 1064 nm, as most particles in the small size range do not scatter enough light to be detected and the detected signal of particles at large sizes is noisy due to limited number concentration (Liu, D. et al., 2014; Taylor et al., 2015). Due to this limited efficiency in the detection range for the scattering channel, the MR vs. $M_{BC}$ distribution was corrected for the size-dependent detection efficiency of the SP2 instrument, following the methods described by Taylor et al. (2015, 2020).

The above SP2 analysis required assumptions of densities and refractive index of both the BC core and the coating. A BC density of 1.8 g cm$^{-3}$ and a BC refractive index ($m_{BC}$) of 2.26-1.26i were used in this study, since these values have been shown previously to provide good agreement with measurements for externally mixed BC particles for scattering at 1064 nm using the Mie core/shell model (e.g. Moteki et al., 2010; Taylor et al., 2015). The coating density was assumed to be the same as bulk non-BC components and was calculated from the AMS measured components following volume mixing rules.

The densities of 1.27 g cm$^{-3}$ for OA and 1.77 g cm$^{-3}$ for inorganic components were applied (Cross et al., 2007). In our optical modelling, we assumed a non-absorbing coating which does not contribute directly to absorption. The refractive index of the coating was assumed to be 1.5–0i, as used in previous works (e.g. Taylor et al., 2015; Liu et al., 2017; Wu et al., 2018). The optical models therefore do not account for direct absorption by non-BC species such as BrC.

Applying the generated 2-D distribution of MR vs. $M_{BC}$ as inputs, different optical models were used to simulate

ensemble mean absorbing properties. We employed the core/shell Mie model, and also several parameterisations which are based on empirical fits to the bulk MAC or absorption enhancement ($E_{Abs}$) for BC particles of different mixing states (Liu et al., 2017; Chakrabarty and Heinson, 2018; Wu et al., 2018). These latter empirical models were chosen based on a previous study using the same optical simulations (Taylor et al., 2020), as they produced MAC and AAE values of aged BB aerosols in relatively good agreement with measurements. We provide details on these optical models and parameterisations in Sect.

S2.2 of the Supplementary Information. For each model, we generated 2-D tables of absorption cross-section or $E_{Abs}$ following their optical schemes, corresponding to the same grid of the 2-D distribution of MR vs. $M_{BC}$ generated from the measurement data. The modelled MACs (at 405, 514 and 658 nm) were determined by the ratio of the integrated absorption cross-section to the total BC mass, or by multiplying the modelled $E_{Abs}$ by the MAC value for uncoated BC recommended by Bond and Bergstrom (2006). The modelled $E_{Abs}$ was determined as the ratio of the simulated bulk absorption cross-section

for coated BC to that for uncoated BC. Predictions of $E_{Abs}$ are output from the core/shell Mie model, in addition to constituting the sole output from the empirical optical parameterisations. The AAE between two wavelengths was determined by Eqn. (1), using the modelled MAC instead of $B_{Abs}$. Taylor et al. (2020) assessed the uncertainties in calculated values for MAC and AAE from these different optical models using a Monte Carlo analysis, which considered the



uncertainties from different input variables (BC mass, MR and non-refractory material concentrations). The derived
uncertainties from Taylor et al. (2020) were considered in this study.

## 3 Results

### 3.1 Sampled fire plumes

On 1 March 2017, the ARA (flight C005) flew over some selected MODIS-detected fires repeatedly (Fig. 1b) and
sampled fresh plumes at different heights (~400 – 1500 m) during the plume rise stage. We assume that the fresh plumes
(commonly referred to *sources*), sampled by positioning the ARA directly over active fires, were less than 0.5 h old. The
emitted plumes were transported by north-easterly prevailing winds. Vertical profiles of measured horizontal winds are
shown in Fig. S1. The aircraft also sampled air immediately downwind from the fires, making plume transects (Fig. 1b). The
downwind plumes had undergone further ~1 h transport and are denoted as near-source. The plume age was estimated by the
distance from the fires and the average wind speed measured by the aircraft. Later the same day, flight C006 sampled
transported smoke as it moved west over the Atlantic Ocean (Fig. 1c). On the following day (2 March 2017), flight C007
sampled transported smoke over the Atlantic Ocean again (Fig. 1d). We selected smoke plumes that were sampled over the
Atlantic Ocean and NAME back trajectories showed that these plumes in flights C006 and C007 were mainly transported
from a similar fire region to that associated with sampling in flight C005. The back trajectories and MODIS-detected fire
indicate that the transport times of selected smoke plumes over the ocean were ~3–6 h in C006 (Fig. 1c) and ~9–12 h in
C007 (Fig. 1d).

Key information regarding the sampled smoke plumes is provided in Table 1. Modified combustion efficiency (MCE)
is widely used to indicate the combustion condition of a fire. An MCE of 0.9 represents a fire comprised of approximately
half-flaming and half-smouldering combustion, and MCE values closer to 1 indicate a higher contribution from flaming
combustion (Reid et al., 2005). Detailed calculation methods for MCE are listed in Sect. S1.1 of the Supplementary
Information. In this study, the calculated MCEs (Table 1) of selected smoke plumes at different transport ages were in a
small range of 0.94 to 0.96, suggesting that all selected smoke plumes during MOYA-2017 were consistently dominated by
flaming-phase combustion emissions. Although the selected smoke plumes are unlikely to be emitted from the same fire at
the same emission time, they originated from similar fire areas that likely have the same fuel type. The fire areas are mainly
a mixture of cropland (agricultural stubble) and wooded savannah (Roberts et al., 2009). Both the similar fuel and MCE
indicate that the selected smoke plumes are likely to be comparable in terms of the initial aerosol properties at source. Table
1 shows that the sampled smoke plumes were warm and dry. The sampled smoke over the Atlantic Ocean was above the
marine boundary layer, mitigating against interference between BB aerosols and the marine environment during plume
transport. Furthermore, there was no precipitation during the selected flights, and thus wet removal of aerosols during the
plume transport is expected to be negligible. Overall, these selected smoke plumes had a discrete range in plume age, from
about <0.5 to ~12 h, which provides an opportunity to study the evolution of BB aerosol properties during the first ~12 h of





transport. In the following sections, we analyse the chemical properties, size distributions, BC core sizes and mixing states, and light absorption properties of submicron BB aerosols in these selected smoke plumes with different ages.

## 3.2 Initial aerosol composition and chemical evolution

In this section, we study the chemical properties of submicron BB aerosols in the fresh plumes and their evolution with
transport time. The submicron aerosol mass concentration and chemical composition fractions were calculated from the measured mass concentrations of species from the AMS and SP2. For composition calculations, the 1-Hz SP2 data were averaged to the AMS sampling rates. We also investigate the fire emission conditions of OA and BC and post-emission OA chemistry. Time series of the concentrations of different chemical components in each flight are shown in Fig. S2.

### 3.2.1 Fresh biomass burning chemical properties

In the fresh plumes (<0.5 h), the mean mass fractions (with standard deviation) of submicron BB aerosols were estimated to be 72% ($\pm$ 5%) for OA, 15% ($\pm$ 6%) for BC, 2% ($\pm$ 0.4%) for nitrate, 0.3% ($\pm$ 0.3%) for sulfate, 5% ($\pm$ 1%) for ammonium and 6% ($\pm$ 2%) for chloride (Fig. 2a). Compared with previously observed fresh smoke aerosol from flaming BB over southern Africa (Vakkari et al., 2014), OA and BC are consistently the largest two contributors to total aerosol mass, making up over 85% of the submicron mass loading. A large fraction of non-refractory chloride was observed in fresh BB
aerosols in our sampling area, which is possibly due to the usage of organochlorine pesticides in Senegal (Diop et al., 2019).

Fire emission information of a species can be represented in two forms: enhancement ratio (ER) and emission factor (EF). In this study, the ERs of BC and OA with respect to CO ($\Delta BC/\Delta CO$ and $\Delta OA/\Delta CO$) were calculated for sampled fresh plumes (< 0.5 h), by dividing the excess BC or OA by the excess concentration of CO. The background concentrations of different species for freshly emitted plumes were determined immediately before entry into and after exiting out of the
plume. The EF of a species is defined as the mass of the species emitted (in grams) with per kilogram of dry matter burnt (Andreae and Merlet, 2001). Details of the calculation methods for ER and EF in fresh plumes are provided in Sect. S1.1 of the Supplementary Information. Although we sampled the fresh plumes over same fires repeatedly in an hour and the MCEs of source fires were similar (0.94 – 0.96), the $\Delta BC/\Delta CO$ and $\Delta OA/\Delta CO$ ratios ($\mu g\ cm^{-3} / \mu g\ cm^{-3}$) in the fresh plumes varied over the ranges of 0.012 – 0.021 and 0.045 – 0.101, with averages of 0.016 and 0.071 respectively. The EFs of BC ($EF_{BC}$)
were also variable, with values recorded over the range of 0.25 – 0.49 g kg$^{-1}$ and with an average value of 0.37 $\pm$ 0.07 g kg$^{-1}$, which is within the range of 0.26 – 0.61 g kg$^{-1}$ reported by previous studies for African savannah (Andreae, 2019). The variations in emission factors indicate that there were temporal fluctuations in the source aerosol emission strength over the same fires in this region.

We converted OA mass into OC mass using the OM/OC ratios estimated from the AMS measurements. In sampled
fresh plumes, the linear Pearson's correlation coefficients between OA and BC ($\rho_{OA-BC}$), and between OC and BC ($\rho_{OC-BC}$) were calculated as 0.87 and 0.88 respectively. The OA and OC mass were found to have positive relationships with BC at source. Using the unconstrained linear orthogonal distance regression (ODR) fitting method for the regression of the mass





concentrations of OA and BC, and OC and BC, we estimated the ΔOA/ΔBC and ΔOC/ΔBC ratios as 7.2 (± 0.9) and 5.0 (± 0.6) respectively. Although the previously estimated aerosol emission factors (ΔBC/ΔCO, ΔOA/ΔCO and $EF_{BC}$) showed temporal fluctuations at source, the ΔOA/ΔBC and ΔOC/ΔBC ratios demonstrated less variance and are likely to be representative parameters for describing aerosol emissions from fire sources during MOYA-2017 period.

### 3.2.2 The evolution of chemical properties

Fig. 2a shows the chemically speciated mass fractions of submicron BB aerosols at different plume ages. Some inorganic (nitrate and sulfate) mass fractions of aerosols were enhanced during the first ~12 h of transport. This observation is consistent with the secondary processing of $NO_x$ and $SO_2$ emitted from BB (Pratt et al., 2011; Akagi et al., 2012). The nitrate mass fraction was also observed to increase and stabilize more rapidly than sulfate, due to the faster transformation of $NO_2$ than $SO_2$ by reaction with OH radicals at typical atmospheric concentrations (Seinfeld and Pandis, 2016). Non-refractory chloride mass fraction showed a decreasing trend with ageing, likely caused by the replacement by other anions such as nitrates (Akagi et al., 2012). During half-day transport, ammonium and BC mass fractions were relatively stable (within their measurement uncertainties) at different ages. Meanwhile, OA constituted a similar fraction of total aerosol mass at different ages but varied in organic composition.

Here, we use some important OA fragment markers (m/z 43, 44, 57, 60) from the AMS to investigate the evolution of organic composition. The ion peak at m/z 60 is attributed to levoglucosan-like species, which has been accepted as a marker of BB pyrolysis products (Schneider et al., 2006). The m/z 43 and 57 markers are from the fragments of saturated hydrocarbon compounds and long alkyl chains and are good indicators of fresh aerosols (Alfarra et al., 2007). The m/z 43 marker can also come from oxidized functionalities such as aldehydes and ketones (Alfarra et al., 2007). The m/z 44 is the signal of $CO_2^+$ ion from carboxylic acid groups and organo-peroxides and suggests the presence of oxygenated organic compounds (Aiken et al., 2008). Fig. 2b shows the $f44$ vs. $f43$ and $f44$ vs. $f60$ diagrams of sampled smoke aerosols at different ages, following the methods in Ng et al. (2010) and Cubison et al. (2011). The $f60$ decreased rapidly in the first 1 h and was close to the background value (0.3 %, Cubison et al., 2011) in environments not influenced by BB after half-day transport. The $f43$ also decreased with transport time. The m/z 57 marker represented 3.7 ± 0.2 % of the total OA mass in the fresh plumes and was below the detection limit after half-day transport. The decreasing $f43$, $f57$ and $f60$ were associated with the substantial decay of levoglucosan-like species and other related primary OA, due to a combination of dilution-driven evaporation and oxidation processes after emission. The $f44$ showed an increasing trend with plume age and indicates an enhanced fraction of oxidized OA or OA in a higher oxidation state in the aged BB aerosols; the increased oxidation state and fraction of OA over particle lifetime is associated with the oxidation of primary OA and secondary organic formation (Ng et al., 2010). The increasing $f44$ has previously been shown to be correlated with a decreasing OA volatility, as the oxygenated OA is less volatile than primary BB OA (Cappa and Jimenez, 2010). Ng et al. (2010) classified OA components into different volatility types based on the $f44$ range, as indicated in Fig 2b. By their method, the OA was mainly semi-volatile oxidized organic aerosol in the first ~1 h and was entirely composed of low-volatility oxidized organic aerosol after





half-day transport. We also calculated the O:C ratio, which is a proxy for OA oxidation state (Aiken et al., 2008). As seen in Fig. 2a (right hand axis), the average O:C ratios (with standard deviation) increased from 0.26 ($\pm$ 0.02) in the fresh plumes to 0.74 ($\pm$ 0.03) at an aerosol age of ~9–12 h, further evidencing the more oxidized OA state of aged BB aerosols.

Previous studies employed ERs to remove dilution effects and quantify post-emission processes within the plume, assuming similar emission conditions at fire source (e.g. Akagi et al., 2012; Jolleys et al., 2012). We calculated the $\Delta BC/\Delta CO$ and $\Delta OA/\Delta CO$ ratios for near-source (~1 h) and transported smoke (~3–12 h; ~9–12 h), using the unconstrained linear ODR fitting between two variables (BC and CO, OA and CO) (Table 2), which is described in more detail in Sect. S1.1 of the Supplementary Information. BC is a chemically stable species. The average $\Delta BC/\Delta CO$ ratios were estimated to be relatively similar within measurement uncertainty at different plume ages, indicating that there was negligible aerosol

removal during plume transport in this study. OA has undergone chemical processing after emission, as indicated by the varying OA compositions described above. Here, we use the features of $\Delta OA/\Delta CO$, $\Delta OA/\Delta BC$ and $\Delta OC/\Delta BC$ ratios to study post-emission OA chemistry. The $\Delta OA/\Delta BC$ and $\Delta OC/\Delta BC$ ratios for near-source (~1 h) and transported smoke (~9–12 h) were also estimated from the unconstrained linear ODR fitting between the mass concentrations for OA and BC, and OC and BC respectively (Table 2). With negligible removal during transport, these ratios would be mainly affected by OA

transformation: the dilution-driven evaporation of OA to the gas phase followed by subsequent oxidation and re-condensation, and also the formation of secondary OA from directly emitted precursor gases (Grieshop et al., 2009; Palm et al., 2020). The decreasing $\Delta OC/\Delta BC$ ratios with transport time suggests that there was a continuous net loss of OC mass during the ageing process, implying that the evaporation loss of OC dominated over condensational growth. The average $\Delta OA/\Delta CO$ and $\Delta OA/\Delta BC$ ratios showed small changes at different ages. The relatively constant $\Delta OA/\Delta CO$ and $\Delta OA/\Delta BC$

ratios during transport arise from the balance of dilution driven evaporation of OC and the increasing O:C ratios for OA, as has been observed previously (e.g. Capes et al., 2008; Pratt et al., 2011).

### 3.3 The evolution of smoke aerosol size

   Fig. 3a shows the mean size distributions of sampled smoke aerosols from the SMPS measurements and the corresponding best-fit lognormal distributions, providing determinations of count median diameter (CMD). It was not

possible to obtain a size distribution in the fresh plumes (<0.5 h) since there was not enough time for the SMPS to obtain a full 1-min scan in the plume transect (<30 sec). We mainly detected single dominant modes during the transects at different ages, transferring from the Aitken mode to the larger accumulation mode. The CMD increased from 85 nm at ~1 h aerosol age to 123 nm after half-day transport. A previous West African BB study observed near-source aerosol size using the PCASP (Passive Cavity Aerosol Spectrometer, optical sizing), and reported a similar CMD of 110 nm (Capes et al., 2008).

Previous measurements of southern African BB aerosols using a DMPS (Differential Mobility Particle Sizer) reported CMDs of 69 nm at an aerosol age of < 0.5 h and grew to 123 nm at ~3 h (Vakkari et al., 2018), which is similar to the growth of mobility particle size measured in this study.



From the SP2 measurements, BC mass was converted to spherical-equivalent BC core diameter ($D_C$) with an assumed BC density of 1.8 g cm$^{-3}$. Fig. 3b shows the mean sphere-equivalent BC core size distributions in sampled smoke plumes, in terms of number and mass. The average BC core mass and number size distributions were similar at different aerosol ages. The BC core CMD and mass median diameter (MMD) were relatively constant during half-day transport, falling in the ranges (10 to 90 percentile) of (100 – 116 nm) and (181 – 207 nm) respectively. The BC core sizes in this study are also similar to the reported mean values (CMD = 121 nm and MMD = 188 nm) for highly aged BB aerosols (>7 days) from flaming burning in southern Africa (Taylor et al., 2020). The relatively stable BC core size at different ages indicates that BC-BC coagulation events are likely to be minor after emission.

Coating thicknesses on BC were also calculated for BC-containing particles from the SP2 measurements, in the $D_C$ range of 110 – 315 nm. This range was determined using the method outlined by Taylor et al. (2015). Fig. 4 shows the measured distributions of single-particle coating properties in selected smoke plumes at different ages, expressed in terms of shell/core ratios and absolute coating thickness. In the fresh plumes (<0.5 h), ~ 40% of BC particles had measurable coatings. BC was dominantly externally mixed with other co-emitted particles at source (<0.5 h). After emission, BC gradually became internally mixed with other species, which condensed materials onto the BC cores. Nearly all BC particles had measured coatings after half-day transport. During this process, BC showed enhanced coating thickness with transport time. The median BC shell/core ratios and absolute coating thickness increased from 1.1 and 13 nm in the fresh plumes (<0.5 h) to 1.7 and 50 nm respectively after half-day transport.

We now summarise the evolving mixing state of BB aerosols in the sampled wildfire plumes. The BB aerosols in fresh plumes (<0.5 h) exhibit a high level of external mixing. Then, the condensation processes between particles would occur after emission, i.e., organic species repartition between particles over particle lifetime. At early plume ages (fresh and near-source), the OA was consistent with that of semi-volatile organic aerosol. There is a dynamic equilibrium of the semi-volatile organic species in OA through evaporation and re-condensation until OA is highly oxidized to form low volatility organic species. Given the higher O:C ratios with transport time (Fig. 2a, and as described in Sect. 3.2), the OA was observed to be highly oxidized and was in the low-volatility range after half-day transport. These lower volatility species would preferentially partition to the particle phase. Furthermore, the inorganic species (nitrate and sulfate) formed from the oxidation of emitted gaseous NO$_x$ and SO$_2$ after emission would also condense onto existing particles. The lower-volatility OA, as well as the formation of inorganic species with transport, contributed to the increasing bulk aerosol CMD and coating thickness on BC.

## 3.4 The evolution of aerosol absorption

### 3.4.1 Absorption parameters

The AAE is an important optical parameter for aerosol characterization and apportionment. For the purposes of this paper, BC absorption is considered independent of wavelength and therefore represented by an AAE of ~1 (Bond et al.,





2013). It is generally assumed that an AAE significantly greater than 1 indicates the presence of non-BC absorbing particles like BrC or dust which have higher AAEs than fresh BC (Lack and Langridge, 2013). In this study, the $AAE_{405-658}$ and $AAE_{514-658}$ (Fig. 5) were ~ 1.1 and 0.9 respectively in the fresh plumes, and both showed increasing trends during the ageing process, reaching up to >2.1 and >1.7 respectively after half-day transport. As an impactor prior to the PAS removed particles with aerodynamic diameters > 1.3 µm, and because Saharan dust is mainly in the coarse-mode size range (e.g.

Ryder et al., 2018), the impact of mineral dust on our AAE measurements should be minor. The AAE may also be affected by changes of BC size and coating thickness with ageing. Numerical optical simulations show that the $AAE_{(300 - 1000 \text{ nm})}$ of fresh and uncoated BC is approximately 1.05 and relatively insensitive to particle size (Liu et al., 2018). For BC particles with core diameters larger than 0.12 µm, the $AAE_{(300 - 1000 \text{ nm})}$ becomes smaller when BC particles are aged due to compaction of structures and the addition of non-absorbing coating materials (Liu et al., 2018). Zhang et al. (2020) also

conducted numerical studies of the $AAE_{(350 - 700 \text{ nm})}$ of polydisperse BC aggregates, using a range of BC core sizes (geometric mean radius: 50 – 150 nm) and coatings (shell/core ratios: 1.1 – 2.7). Calculations by Zhang et al. (2020) indicate that the $AAE_{(350 - 700 \text{ nm})}$ of clear-coated BC is slightly sensitive to particle microphysics (e.g. BC core size and shell/core ratios), with values mainly in the range of 0.7 – 1.4. However, when BC is coated with absorbing material, such as BrC, numerical optical simulations show larger AAEs than clear-coated BC, and also show an increasing AAE with enhanced coating

thickness (Gyawali et al., 2009; Lack and Cappa, 2010; Zhang et al., 2020). Some theoretical calculations suggest that a threshold value of AAE > 1.6 is strongly indicative that BC coatings contain light absorbing materials (e.g. Gyawali et al., 2009; Lack and Cappa, 2010). The evolution of BC size and coatings with ageing, as described in Sect. 3.3, are unlikely to dominate the change of observed AAE if there is no BrC. The above discussions indicate that the observed increasing AAE is most likely attributable to the formation of BrC. In this study, BC was the dominant light-absorbing aerosol in fresh BB

plumes, while BrC was likely a minor component at source but was formed during transport.

The MAC ($m^2$ $g^{-1}$) is a key variable for characterising the absorbing properties of aerosols. The measured aerosol absorption normalized to BC mass, denoted as the $MAC_{\text{meas-BC}}$, was determined by the unconstrained linear ODR fitting between the measured absorption coefficient and BC mass concentration for near-source (~1 h) and transported smoke (~ 3 – 6 h; ~ 9 – 12 h). The measured 1-Hz absorption and BC mass concentration in smoke were averaged to 10-s sampling

periods to lower the uncertainty introduced by small differences in instrument response time. Bond and Bergstrom (2006) reported a MAC value of 7.5 $m^2$ $g^{-1}$ at $\lambda$ = 550 nm and assumed an AAE of 1 for fresh and uncoated BC, which could be used to extrapolate the MAC of uncoated BC to different wavelengths. The absorption enhancement ($E_{\text{Abs-MAC}}$) was then calculated by the ratio of the $MAC_{\text{meas-BC}}$ to the MAC of uncoated BC derived from Bond and Bergstrom (2006). $E_{\text{Abs-MAC}}$ represents the additional absorption of light above that expected from the bare BC cores and is attributed to the lensing effect

of coatings on BC cores and the absorption by BrC (Cappa et al., 2012; Lack et al., 2013). Fig. 6 shows the calculated $MAC_{\text{meas-BC}}$ and $E_{\text{Abs-MAC}}$ at 405, 514 and 658 nm for aerosols in sampled smoke at different ages, demonstrating consistent increases in these metrics with particle age. The previous section has shown that coating thicknesses of BC-containing particles increased with particle age. The absorption enhancement from the lensing effect of non-absorbing coatings is





expected to be nearly identical over a broad spectral range (e.g. Cappa et al., 2012; Nakayama et al., 2014; Pokhrel et al.,

2017). However, the formation of BrC as indicated by our measured AAE values resulted in higher observed $E_{Abs-MAC}$ for progressively shorter wavelengths.

### 3.4.2 Absorption attribution to BrC

In this section, we attribute aerosol absorption to BrC using different methods. The calculations were based on both the measurements described in previous sections and simulated absorption properties derived from optical models. We used the

measured $MAC_{meas-BC}$ shown in Sect. 3.4.1 and the modelled MAC of aged BC to attribute aerosol absorption. We also used the AAE method in Lack and Langridge (2013), as described in Sect. S1.2 of the Supplementary Information, to estimate BrC absorption contribution. The modelled AAEs of aged BC were used in this AAE attribution method.

**1)  Modelled absorbing properties of aged BC with non-absorbing coatings**

In this section, we present the modelled mean MAC and AAE of aged BC with non-absorbing coatings (clear-coated

BC), for selected smoke at different ages. Firstly, we determined the size and mixing state of BC-containing particles, following the processes in Sect. 2.3. Fig. S3 shows the 2-D distribution of BC mass and mixing state (MR vs. $M_{BC}$) at different aerosol ages. For BC-containing particles in selected smoke at ~1 h and ~3–6 h ages, the distributions were similar, with the MR centred around 2 – 4. After half-day transport, the MR distribution was centred around 5 – 7 for a BC core mass of ~ 1 femtogram (fg), and the MR decreased for larger core sizes. These 2-D distributions of MR vs. $M_{BC}$ provided the

required information for input into different optical models to predict ensemble mean MAC and AAE for clear-coated BC. Sect. S2 of the Supplementary Information provides further details on the different optical models and parameterisations used here. The calculations based on the core/shell Mie model were termed "CS". We also calculated the absorption enhancement "$E_{Abs}$" from the core/shell Mie model; these calculations were termed "CS-$E_{Abs}$". The CS-$E_{abs}$ method determines "$E_{Abs}$" as the ratio of the simulated bulk absorption cross-section for clear-coated BC to that for uncoated BC

from the CS method. The MAC of clear-coated BC was then calculated by multiplying the modelled $E_{Abs}$ and the MAC of uncoated BC ($MAC_{BC}$) from Bond and Bergstrom (2006). This CS-$E_{Abs}$ method corrects the MAC for clear-coated BC using $MAC_{BC}$ values (7.5 $m^2$ $g^{-1}$ at $\lambda$ = 550 nm, with AAE = 1) that are summarised from previous literatures and are commonly accepted as best estimates. We also considered various values of the refractive index of BC ($m_{BC}$) in core/shell Mie models, as listed in Table S1. The calculations based on empirical fits to the $E_{Abs}$ or MAC from Liu et al. (2017), Chakrabarty and

Heinson (2018) and Wu et al. (2018) were termed "Liu-$E_{Abs}$", "Chak-MAC / Chak-$E_{Abs}$" and "Wu-$E_{Abs}$" respectively.

Fig. S4 shows the modelled mean values of MAC at 405, 514 and 658 nm for clear-coated BC as a function of the imaginary component of the $m_{BC}$ ($k_{BC}$) at different plume ages. Simulated MAC values varied between different optical models. It should be noted that, the modelled MACs at 405 and 514 nm from the CS method were not considered in the following analysis, as Taylor et al. (2020) has discussed the underprediction of MAC at short wavelengths from the

core/shell Mie model. In Mie models, the intensity of light decreases when penetrating through an absorbing sphere, since the surface of the sphere would absorb light and shield the centre. For small particles, this shielding effect is small, but for





large particles, the centre of a spherical particle is effectively shielded from exposure to light. In reality, BC is a non-spherical fractal aggregate with a porous structure and a high surface-to-volume ratio. This high surface area relative to the total BC mass allows light to fully interact with the BC component and the shielding effect is diminished (e.g. Chakrabarty

and Heinson, 2018). Therefore, the shielding effect in Mie models leads to an underestimation of light absorption for the BC particles. Fig. S5 in Taylor et al. (2020) has shown that this shielding effect underestimates the MACs at λ = 405 and 514 nm but not for λ = 658 nm for BC particles with core sizes in the range ~ 150 – 200 nm, and we note that the MMD of BC cores measured in this study were also in this size range. In addition, the MACs from the CS-$E_{Abs}$ method were neglected for particles at ages of ~1 h and ~3–6 h. At these early lifetimes when the MRs of BC-containing particles were low, the

core/shell Mie model would overestimate the $E_{Abs}$ because non-BC components are unlikely to form a shell surrounding the BC, but rather fill internal voids in the porous soot structure (e.g. Liu et al., 2016; Kahnert et al., 2017; Pei et al., 2018). We also excluded the modelled MACs at 405 and 514 nm from "CS-$E_{Abs}$" and the modelled MACs at 658 nm from "CS" and "CS-$E_{Abs}$" using a complex $m_{BC}$ of 2.26-1.26i, since these values exceeded the experimental uncertainty of the measured MAC of coated BC in Taylor et al. (2020).

Fig. S5 summarises the reasonable ranges for modelled MACs for clear-coated BC in selected smoke with different ages. Generally, the "Wu-$E_{Abs}$" scheme provided the lowest values in these modelled MACs, while the "CS-$E_{Abs}$" or "Chak-$E_{Abs}$" scheme gave highest values. "Liu-$E_{Abs}$" gave approximately middle estimates among these schemes. The modelled MACs of clear-coated BC at our three wavelengths demonstrated the same increasing trends with transport time. For example, the modelled mean $MAC_{405}$ at ~1 h was in a range of 12.0 – 14.2 $m^2$ $g^{-1}$, slightly increased to 12.2 – 14.4 $m^2$ $g^{-1}$ at

~3–6 h, and increased further to 13.4 – 16.4 $m^2$ $g^{-1}$ at ~9–12 h.

Fig. S6 shows the modelled mean values of $AAE_{405-658}$ and $AAE_{514-658}$ for clear-coated BC in selected smoke plumes. Simulated AAEs were similar between different optical models, except the values derived from the "CS" method. The underprediction of absorption by the core-shell Mie model at short wavelengths due to the shielding effects leads to unrealistically low AAE values, and thus the modelled AAEs derived from the "CS" method were excluded in the following

analysis. The "Chak-MAC / Chak-$E_{Abs}$" schemes fixed AAE at exactly 1. The "CS-$E_{Abs}$" and parameterisations from "Liu-$E_{Abs}$" and "Wu-$E_{Abs}$" used modelled $E_{Abs}$ multiplied by the MAC of uncoated BC from Bond and Bergstrom (2006) to calculate the MAC of clear-coated BC. Given that Bond and Bergstrom (2006) assumed the AAE for uncoated BC is 1 and that the $E_{Abs}$ caused by clear coatings is approximately constant over the visible spectrum (e.g. Cappa et al., 2012; Nakayama et al., 2014; Pokhrel et al., 2017), it is unsurprising (given the modelling details provided above) that the "CS-$E_{Abs}$", "Liu-

$E_{Abs}$" and "Wu-$E_{Abs}$" schemes all gave AAEs near 1. Moreover, these predictions are in agreement with previous numerical studies that also demonstrated the AAE of clear-coated BC as near 1.0 (e.g. Liu et al., 2018; Zhang et al., 2020). We summarise by stating that our modelled AAEs (excluding "CS") for clear-coated BC at different ages are within the limited range of 1.0 – 1.1, and the modelled AAEs show minor changes with transport time.

**2)  Absorption attribution**





With the ranges of modelled MAC and AAE for clear-coated BC derived from the previous section, we attributed observed aerosol absorption into BrC using two methods. The first method is using the measured $MAC_{meas-BC}$ described in Sect. 3.4.1 and the modelled MAC ($MAC_{modelled}$) values. Fig. 6 shows the measured $MAC_{meas-BC}$ and the lowest, middle and highest $MAC_{modelled}$ at wavelengths of 405, 514, and 658 nm. The MAC of uncoated BC reported by Bond and Bergstrom (2006) is also included in Fig. 6. Here, differences between the MAC of uncoated and clear-coated BC represent absorption

enhancement due to the lensing effect caused by non-absorbing coatings. Further differences between the $MAC_{modelled}$ of clear-coated BC and the measured $MAC_{meas-BC}$ are attributed to BrC absorption. We calculated the range of BrC contribution to total aerosol absorption, using these $MAC_{modelled}$ and measured $MAC_{meas-BC}$ [($MAC_{meas-BC}-MAC_{modelled}$) / $MAC_{meas-BC}$]. As seen in Fig. 7, the estimated BrC contribution using this method varies considerably and is enhanced with ageing. The fractional contribution of BrC at 405 nm was in a range of 7 – 22 % at ~1 h age and increased to 18 – 33 % after half-day

transport. The estimated BrC absorption fraction at 514 nm was lower than that at 405 nm and was negligible at early aerosol ages (~1 h; 0 – 15 %) but significant after ~9–12 h of aerosol ageing (10 – 26 %). The BrC absorption fraction at 658 nm was estimated to be smallest among three wavelengths, increasing from a range of 0 – 14 % at ~1 h to 5 – 23 % at ~9–12 h.

From the first method, the upper bounds of BrC contribution at 658 nm calculated using the lowest $MAC_{modelled}$, were ~ 20%. However, previous studies have observed that BrC absorption decreases significantly from near-UV to visible ranges

and is negligible close to the wavelengths of 700 nm (e.g. Laskin et al., 2010; Liu et al., 2015). The contribution of BrC to total aerosol absorption is therefore expected to be negligible at 658 nm. In this method, the lowest $MAC_{modelled}$ calculated from the "Wu-$E_{Abs}$" scheme is likely to be underpredicted and leads to overestimated upper bounds of BrC contribution fraction. The low bounds of the estimated contribution fraction at 658 nm were minor throughout the transport time and are therefore likely to be more representative than the upper bound estimate.

Fig. 7 also shows our estimates of BrC fractional contribution to total aerosol absorption predicted using the AAE method described by Lack and Langridge (2013). In brief, this method assumes no absorption contribution from non-BC species at 658 nm and extrapolates BC absorption from 658 nm to shorter wavelengths using the AAE value of BC ($AAE_{BC}$), and then calculates the BrC absorption by subtracting BC absorption from the total aerosol absorption. More details are described in Sect. S1.2 of the Supplementary Information. A key determinant of predictions using this method is the choice

of $AAE_{BC}$. We used the range of modelled AAEs ($AAE_{405-658}$ and $AAE_{514-658}$) for clear-coated BC determined in the last section that were in the range of 1.0 – 1.1. The estimated BrC contribution fractions from this method were in a small range, as the modelled AAEs were all near 1. The fractional contribution of BrC to total aerosol absorption at 405 nm was in the range of 15 – 18 % at ~1 h age and increased to 28 – 31 % after half-day transport. The uncertainty of this AAE attribution method is significant when the fractional contribution of BrC to aerosol absorption is low (Lack and Langridge, 2013). Thus,

the estimated low BrC contribution at 514 nm is likely to be inaccurate here. As we have assumed no absorption contribution from non-BC species at 658 nm, the BrC contribution fraction at 658 nm was constrained to 0.

From above two methods, the estimated BrC absorption contribution both showed strong wavelength dependence, with higher contributions at shorter wavelengths. This is consistent with the stronger BrC absorptivity at shorter wavelengths



from visible to near-UV light range, as reported in the literature (e.g. Feng et al., 2013; Laskin et al., 2015). We calculated

similar BrC contribution fractions at 405 nm, with the uncertainty in calculated BrC fraction from the AAE method largely within the uncertainties from the attribution method of comparing $MAC_{modelled}$ and $MAC_{meas-BC}$. We combined the results from these two calculations, considering that the upper bounds of the BrC attribution from the $MAC_{modelled}$ versus $MAC_{meas-BC}$ are likely overestimated, and the calculated BrC contribution fractions at 405 nm were estimated to be in the range of 18 – 31 % after half-day transport. The ranges of BrC contribution fractions at 514 nm showed large variations between two

methods, which are due to the large uncertainties when the contribution is low. The fractions at 514 nm estimated from the AAE method are close to the low bounds of fraction from the $MAC_{modelled}$ versus $MAC_{meas-BC}$ method. BrC contribution at 514 nm is likely small at early aerosol ages but should be considered (attributed to at least 10 % of the aerosol absorption) after half-day ageing.

We stress that the methods used here have drawbacks. In our optical modelling, we simulated the aged BC with non-

absorbing coatings. However, the absorbing BrC, which is found to be formed upon ageing in this study, will be internally mixed with other (clear) components of coatings on BC. Previous simulations have demonstrated that the $E_{Abs}$ due to the lensing of coatings on BC is reduced when the coating is mildly absorbing (i.e. contains BrC) relative to the enhancement induced by completely non-absorbing coatings (e.g. Lack and Cappa, 2010). If using the assumptions of absorbing coatings and considering this reduction of $E_{Abs}$ in our optical models, it will lead to a higher bound on BrC attribution from the

method of comparing $MAC_{modelled}$ and $MAC_{meas-BC}$. This reduction of $E_{Abs}$ is sensitive to both the thickness and the imaginary refractive index of absorbing coatings (Lack and Cappa, 2010). Some studies have used direct measurements of $E_{Abs}$ (from the ratio of ambient absorption to thermal-denuded absorption) as model constraints to calculate the imaginary refractive index of BrC and also BrC contribution (e.g. Lack et al., 2012b; Pokhrel et al., 2017). However, such direct measurements of $E_{Abs}$ were not performed in this study. As the BrC mass fraction and mixing states of BC evolved with ageing time, the

imaginary refractive index of the absorbing coating would also change during the half-day transport. It is beyond the scope of this work to ascertain a reasonable refractive index of absorbing coatings. More explicit work could be done to investigate the BrC refractive index and its contribution in future studies of West African BB. Furthermore, a recent laboratory study proposed unidentified absorbing particles from BB that do not correspond to BC or BrC (Adler et al., 2019). This new class of species strongly absorbs red light rather than at short wavelengths, possibly with MAC and AAE between that of BrC and

BC. Thus, the absorption may not be simply attributed to BC and BrC. Limited information is known for this class of species and more work is needed to identify how they may affect absorption attribution.





## 4 Discussion

### 4.1 Correlation between evolution of optical and chemical properties

Our results show increasing AAE trends during the half-day transport of smoke aerosols emitted from West African
BB, indicating continuous formation of BrC and its increasing role on total aerosol absorption. As the AAE and BrC
influence increased, the $f44$ and O:C ratio also showed increasing trends while the $f60$ decreased. This suggests that BrC in
this study is poorly related to the primary OA, but closely linked to the oxygenated and low-volatility OA formed during
ageing that is likely from the oxidation and re-condensation of evaporated primary OA and subsequent secondary OA
formation. Laboratory studies provide evidence that the processing of BB OA would increase the BrC absorption efficiency
over the UV-vis range (Li et al., 2020), and secondary OA produced in aged BB can contain BrC and absorb light to a
significant extent (Saleh et al., 2013).

Some organic nitrates are reported to be important contributors to BrC. These can be formed from the high-$NO_x$
photooxidation of polycyclic aromatic hydrocarbons (PAHs) and oxygenated aromatics (e.g. phenols) emitted from BB
(Laskin et al., 2015; Ahern et al., 2019). For example, BrC has been shown to be generated by the OH-oxidation of
naphthalene under high-$NO_x$ conditions in chamber simulations (Lee et al., 2014). Field studies observed that phenolic
compounds and their oxidation products (nitrophenolic products) contribute to BB secondary OA formation and are
substantial contributors to total BrC absorption at 405 nm in wildfire plumes (Mohr et al., 2013, Palm et al., 2020). Although
we cannot separate inorganic and organic nitrates from the C-ToF AMS explicitly, the relationship between ions $NO^+$ (at m/z
30) and $NO_2^+$ (at m/z 46) from the AMS has previously been suggested to be an indicator of nitrate types (Rollins et al.,
2010). The organic nitrates measured by the AMS are reported to have higher m/z 30 to m/z 46 ratios compared to $NH_4NO_3$,
since they decompose further before ionization. The average m/z 30 to 46 ratios were ($3.1 \pm 0.3$) in the fresh plumes, and
became larger and more variable (3–7) in aged smoke, which were several times higher than the average ratio ($1.3 \pm 0.2$)
from the AMS calibration using $NH_4NO_3$ particles. This indicates a contribution from organic-linked nitrate in our reported
total measured nitrate. We estimated the concentrations of organic-linked nitrate following the methods proposed by Farmer
et al. (2010) and modified by Kiendler-Scharr et al. (2016). The detailed methods are described in Sect. S1.3 of the
Supplementary Information. The organic-linked nitrate over total OA mass ratios increased from ($2.0 \pm 0.1$) % in the fresh
plumes to ($4.4 \pm 1.7$) % after half-day ageing. Organic-linked nitrates are likely to have an enhanced fraction in aged BB
OA, which may contribute to the BrC enhancement as the plume ages.

### 4.2 Comparison with highly aged African BB aerosols

Wu et al. (2020) and Taylor et al. (2020) characterized the properties of highly aged BB aerosols from southern African
wildfires, which have undergone >7 days transport after emission. Their derived MCE (~0.97) and ΔBC/ΔCO ratios (0.0087
– 0.0134) are in a similar range to those obtained during MOYA-2017, indicating the same flaming-controlled burning at
source and the biomass fuels (consisting of savannah vegetation and agricultural waste) are broadly similar. Pistone et al.





(2019) also sampled southern African BB aerosols over the Atlantic Ocean and reported optical properties (e.g. AAE) at an

aerosol age of ~4 days. If the results from these studies are combined, a full life picture of BB aerosols emitted from flaming burning of agricultural waste and savannah can be considered. The $f44$ (18 – 23%), bulk aerosol CMD (232 nm) and BC shell/core ratios (2.2 – 2.6) of highly aged BB aerosols in Wu et al. (2020) and Taylor et al. (2020) are all greater than those in this study, indicating the continued chemical ageing of OA and development of particle mixing after half-day transport.

MOYA-2017 suggests that BrC from flaming burning of agricultural waste and savannah initially contributes a minor

fraction of total aerosol absorption but undergoes a net enhancement by photochemical processing within the first 12 h. The $AAE_{470-660}$ of BB aerosols (~4 days) measured by a 3-wavelength PSAP (Particle Soot Absorption Photometer) and reported by Pistone et al. (2019) was in a range of ~1–1.5. The average $AAE_{405-658}$ and $AAE_{514-658}$ of highly aged BB aerosols (>7 days) measured by the PAS were reported to be 1.16 and 0.94 respectively (Taylor et al., 2020). The results of these studies on highly aged BB aerosols indicate that BrC becomes a minor contributor to aerosol absorption again after long-range

transport. After the initial stage of BrC enhancement (the first ~12 h) observed during MOYA-2017, as aerosols continue to age, BrC net loss is expected to occur, coupled with decreasing AAE. A similar evolution was reported in laboratory studies by Wong et al. (2017, 2019). They photolytically aged solution-extracted aerosols generated by wood burning and observed an initial stage of absorption enhancement (~10–20 h) at short visible wavelengths followed by a subsequent decrease of BrC absorption over a longer period (~20–40 h) to below the initial values. A recent chamber study quantified the evolution of

smoke generated by burning various biomass fuels (Cappa et al., 2020). Under one set of test conditions, the BB smoke had initial OA/BC ratios of 3.5–12, which is similar to the initial OA/BC of ~7 observed during MOYA-2017. They measured an increase of AAE at the first stage (< 1 day), in agreement with our field observations. However, the AAE remained relatively constant after ~1-day during their laboratory studies. This difference may be due to the limited light intensity and hence lower photolysis rates when using simulated light in the laboratory studies compared to natural sunlight. Both field and

laboratory studies consistently suggest an initial stage of BrC enhancement after BB emission under certain combustion conditions.

### 4.3 Comparison with BB evolution in other field studies

BB over the Northern Hemispheric part of Africa has regional pattern and characteristics. Every year, areas of cropland mixed with wooded savannah in the sub-Sahelian region experience a short fire season mainly between November to

February (Roberts et al., 2009). As the agricultural residue (stubble) is generally dried after harvest, one feature is that the BB in this area is highly efficient, leading to the consistently high MCE observed in this study. Furthermore, mineral dust is emitted from the Northern Sahara Desert throughout the year and transported to the sub-Sahelian region, and so the BB aerosols are likely to be mixed with dust in different proportions (Johnson et al., 2008). Although our aerosol measurements in this study focused on sub-micron size, which excluded mineral dust, the dust may have impacts during transport of BB

aerosols after emission. For example, the mineral dust can play a role as a condensation sink for BB aerosols. In this section,





we compare this study with previous field observations to investigate the region-specific characteristics of BB aerosol evolution in this area.

      In this study, the characterization of chemical (chemical components, OA composition and oxidation state), physical (bulk aerosol size, BC core size and coating thickness) and optical (AAE and MAC) properties in selected smoke plumes at different ages reveal continuous evolution of submicron BB aerosols during the first half-day transport after emission. Freshly emitted plumes would mix with background air during transport and dilute. Nearby background air out of the plume, which probably consisted of regional haze and aged BB emissions, were also characterized (Table S2). The mass concentrations of different aerosol species and the aerosol absorption in transported smoke at ~1 h and ~3–6 h were up to ~20 times greater than nearby background conditions, suggesting a negligible effect of mixing with background air on aerosol properties in transported smoke at ~1 h and ~3–6 h. Smoke at ~9–12 h were elevated by a factor of ~5 compared to both nearby background aerosol concentration and absorption. Based on the method of Murphy et al. (2009), ~20 % of the observed aerosol at ~9–12 h is likely due to mixing with background aerosol. As the aerosol properties in nearby background air (Table S2) were similar to the smoke aerosols at ~9–12 h, this mixing would not affect smoke aerosol properties significantly. Therefore, the evolution of BB aerosol properties reported in this study is dominated by chemical and physical processing during transport.

      Compared to previous field measurements of BB, this study reports similar chemical and size evolution after emission, regardless of the combustion phase and emission conditions. There are enhanced compositions of some inorganic species and OA oxidation state (Capes et al., 2008; Pratt et al., 2011; Akagi et al., 2012; Kleinman et al., 2020), decreasing proportions of levoglucosan-like species (Cubison et al., 2011; Forrister et al., 2015; Kleinman et al., 2020) and growing bulk aerosol size (Vakkari et al., 2018). This study demonstrates increasing coatings on BC with transport time, consistent with previous field measurements of higher BC coating thickness as they experienced a longer transport period (e.g. Akagi et al., 2012; Perring et al., 2017; Cheng et al., 2018). However, the BC mixing rates with other aerosol components show differences between field studies. In this study, the median BC coating thickness experienced continuous increase, from ~ 13 nm at source (< 0.5 h) to ~ 50 nm after ~12 h. Some observations reported that the majority (> 80%) of measured BC particles have been thickly coated after ~1 h since emission (Akagi et al., 2012) and the BC had a constant thick coating (~100 nm) over an ageing time of ~1–50 h (Forrister et al., 2015). These studies were of smoke from mostly smouldering-controlled fires at source. Compared with these observations, our study with flaming-controlled burning presented a much slower and weaker process of BC mixing with other aerosol components after emission. The differences between studies imply that the fire conditions can play an important role in determining the initial coating thickness and evolution of BC particles. Laboratory experiments have provided some evidence for this, suggesting that BB comprised of higher smouldering combustion fractions generates higher OA/BC ratios and thicker coated BC (Pan et al., 2017; McClure et al., 2020). Furthermore, the co-existing mineral dust in this region may also affect the condensation rate of non-BC material on BC surface, as they may preferentially condense on larger-size dust than the BC, which would lower the process of BC mixing.



The AAE and BrC during MOYA-2017 behaved differently to many previous field studies. Lack et al. (2013) sampled near-source smoke emitted from a large Ponderosa Pine forest fire near Boulder, Colorado. They found that the $AAE_{404-658}$ and non-BC absorption at 404 nm were positively correlated to the $f60/f44$ ratio. Their measured BrC was linked to primary OA, likely the levoglucosan-based chromophores. Forrister et al. (2015) reported the evolution of BrC from western U.S. wildfire emissions, which presented a reduction of $AAE_{532-470}$ dropping from ~4.0 near the fire (~1 h) to ~1.5 after 2-days transport and a net BrC loss with ageing. The decreasing BrC signal correlated well with the increasing O:C ratios and the decreasing $f60$. This case study also indicates a major contribution of primary OA to BrC and suggested that the chemical reaction loss dominated BrC evolution. Wang et al. (2016) investigated the BrC absorptivity in BB plumes emitted from Manaus (Amazon) forest wildfires and found that the MAC of OA decreased with photochemical ageing. Forrister et al. (2015) and Wang et al. (2016) predicted similar half-life time of BrC as ~9–15 h and ~14 h respectively. The case study in Forrister et al. (2015) had smouldering-controlled burning at source, which yielded much higher initial OA/BC ratios (> 100) than the flaming-controlled burning in this study (~7), and gave larger initial AAE, since smouldering burning generally favours the formation of OA rather than BC. It follows that BB emissions with primary compositions dominated by organic matter are more likely to contain more significant BrC than those dominated by BC content (McClure et al., 2020). The forest wildfires in Colorado and Manaus also have different emission conditions compared to MOYA-2017. It is suggested that the opposite behaviours of BrC between these case studies and MOYA-2017 are likely due to the different combustion and emission conditions at fire sources, which gives different initial AAE and BrC contribution between the studies. The varying evolution between studies also implies differences in the dominant processes driving BrC chemical and physical transformation after emission. For BB studies by Forrister et al. (2015) and Wang et al. (2016), the loss of BrC from evaporation and photobleaching by both direct photolysis and OH oxidation is likely to rapidly dominate over any formation process, leading to the decreasing AAE throughout the plume lifetime. For flaming-controlled BB especially with very high BC emissions like in this study, the BrC formation is likely to exceed co-existing loss during at least the first half-day transport, leading to the net BrC enhancement and increasing AAE, after which photobleaching loss is predicted to be the dominant process.

Recent model studies generally assume BrC fractional contribution to OA and optical properties based on laboratory measurements (Feng et al., 2013), and sometimes consider an ageing scheme with the photochemical "whitening" of BrC (Wang et al., 2018; Brown et al., 2018). However, the MOYA-2017 results show that BrC was a small fractional component at source and enhanced with transport. BrC loss is not always the dominant process during ageing in the ambient atmosphere. The formation of BrC cannot be neglected under similar BB conditions to those measured during MOYA-2017.

Overall, comparisons between this study and previous field observations show consistency in the evolution trends of some properties with ageing. However, the comparisons also imply that the life cycles of BC and BrC can vary with burn conditions. Different treatments of BC and BrC properties and their evolution under diverse BB conditions are necessary when modelling regional radiative forcing. Model assumptions constrained by the field data reported here will be useful for





improving predictions of BC and BrC and estimating their radiative effects in BB regions controlled by flaming burning of agricultural waste and savannah-like biomass fuels, i.e. Southern and central Africa, West Africa.

## 5 Conclusions

In the MOYA-2017 campaign, we investigated the evolution of smoke aerosols emitted from flaming-controlled BB in West Africa over the first half-day following emission. Aerosol ageing in plumes from such fires, which are rich in BC emissions, have been rarely reported. These data provide unique and novel field results of absorbing aerosol properties over this important seasonal BB region, using instruments that enable accurate and direct measurements of BC properties and absorption coefficients. Model simulations constrained by these ambient data will be useful for predicting regional radiative effects, specifically in regions affected by flaming-controlled combustion of agricultural waste and savannah-like fuels, such as those in Southern and West Africa.

During the half-day ageing, rapid evolution in chemical and physical properties occurred concurrent with substantial changes in aerosol optical properties (e.g. AAE, MAC). The evolution trends of some properties with age, e.g. enhanced mass fractions of some inorganic species, increasing OA oxidation state and bulk aerosol size, are similar to those observed in previous BB studies. The BC demonstrated increasing internal mixing with other species, quantified using BC shell/core ratios. The effect of ageing on $MAC_{meas-BC}$ (the total aerosol absorption normalized to BC mass), represented by the absorption enhancement ($E_{Abs-MAC}$), was also estimated. $E_{Abs-MAC}$ dramatically increased upon half-day ageing due to the lensing effect of increasingly thick coatings on BC and the absorption of BrC. In this study, BrC contributed only a minor fraction to total aerosol absorption in the fresh plumes (< 0.5 h). Then, an initial stage of BrC net enhancement was observed within the first 12 h after emission as indicated by the increasing AAE with ageing. The generated BrC was found to be positively correlated with oxygenated and low-volatility OA, likely from the oxidation of evaporated primary OA and secondary OA formation. Using different methods, the estimated BrC contribution to total aerosol absorption showed an increasing trend with ageing and was ~ 18–31 % at 405 nm after half-day transport. From comparison to recent field studies reporting African BB aerosol properties over long timescales (>>12 hours), we expect the initial BrC enhancement observed in this work to be followed by a BrC loss-dominant process commensurate with a decreasing AAE. In this region, we observed a different temporal evolution of BrC compared with previous BB studies that mainly focused on emissions from smouldering fires, which have shown a high contribution from BrC at source and BrC net loss upon ageing (e.g. Forrister et al., 2015; Wang et al., 2016). This study demonstrates the importance of BrC formation from secondary processing in West Africa wildfires rather than the primary emissions reported in other BB studies. The varying BrC behaviours between different types of fires indicate that different treatments of smoke aerosol properties and their evolution should be considered when modelling regional radiative forcing.



*Data availability.* Airborne measurements are available from the Centre for Environmental Data Analysis https://catalogue.ceda.ac.uk/uuid/d309a5ab60b04b6c82eca6d006350ae6. The data which are not on this website can be provided by request.

*Author contributions.* G.A. is the PI of this project; H.C. designed the research; J.M.L., P.I.W., M.F., M.I.C., C.F., J.L. and P.B. performed field experiments; H.W., J.W.T, C.Y., J.M.L., K.S., J.D.A. and P.B. prepared datasets of the AMS, SP2, PAS and FGGA; H.W. and J.W.T. performed the optical modelling; H.W. performed NAME back-trajectory analysis. H.W. analysed datasets. H.W. led the manuscript writing and all co-authors contributed to the writing.

*Competing interests.* The authors declare no competing interests.

*Acknowledgements.* This work was funded by the Natural Environment Research Council (NERC) (The Global Methane Budget, University of Manchester reference: NE/N015835/1). The staff of Airtask, Avalon Engineering and FAAM are thanked for their thoroughly professional work, before, during and after the deployment. The NAME group in the UK Met Office are thanked for their instructions on back-trajectory simulations.

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




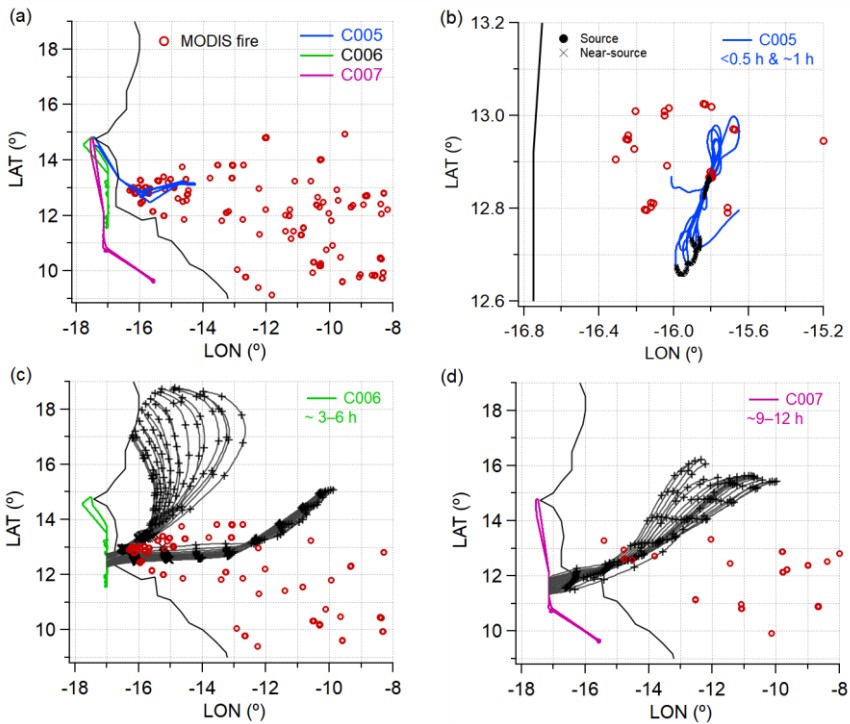

**Figure 1:** (a) Tracks of the flights (labelled from C005 to C007) used in this study, coupled with integrated MODIS-detected fires during campaign days. (b) The selected fresh and near-source plumes, with the spatial distribution of MODIS-detected fires during flight C005. (c-d) 1-day back trajectory of selected sampled smoke over the Atlantic Ocean during flight C006 (c) and C007 (d), marked (black crosses) with every 3h increment. The MODIS-detected fires are also shown in the plots (as observed 3 - 12 h before the sampling period).

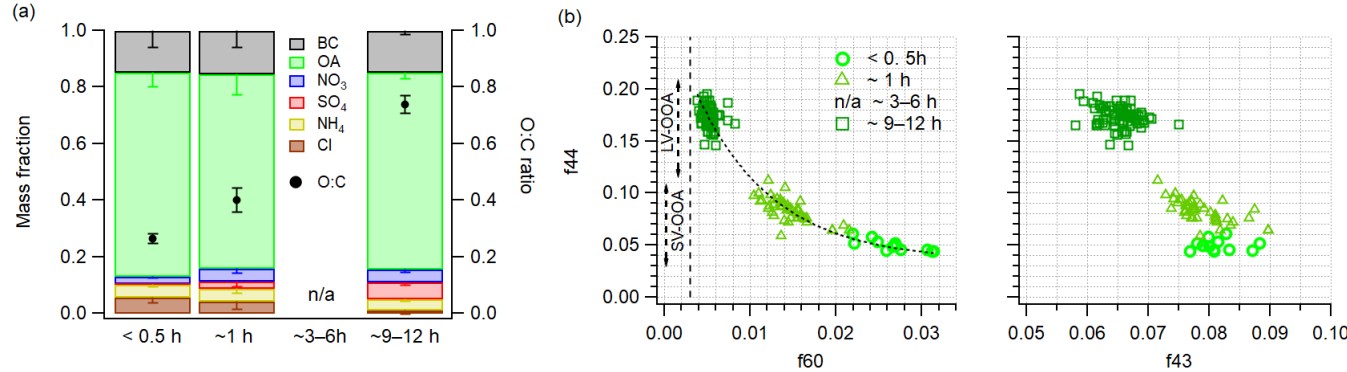

**Figure 2:** (a) The average chemical compositions of sampled smoke submicron aerosols at different ages (left axis), and the black solid circles represent the average O:C ratios of OA in sampled smoke (right axis). The whiskers represent one standard deviation. (b) The fractional signals $f44$ vs. $f60$ and $f44$ vs. $f43$ of sampled smoke aerosols in our study. The dashed vertical line represents the background of $f60$ (0.3%) in environments not influenced by BB, as recommended by Cubison et al. (2011). The dashed-dot line passing through measurement data indicates the general trend in $f44$ vs. $f60$ with aerosol age.

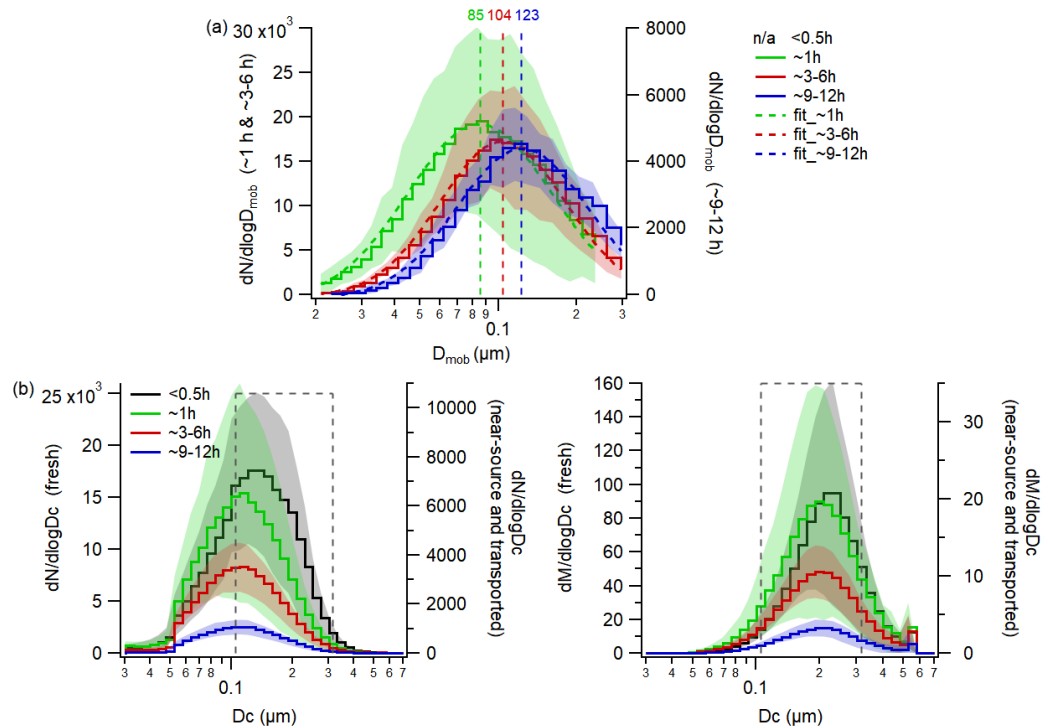

**Figure 3: (a)** The mean number size distributions of sampled smoke aerosols measured from the SMPS. The log normal fitted CMDs are also specified (in nanometres) for aerosols at different ages. The shade areas represent one standard deviation. **(b)** The mean number (left) and mass (right) distributions of the BC core as a function of the sphere-equivalent diameter, for sampled smoke plumes at different ages. The shaded areas represent one standard deviation. The grey dashed square regions show the range of BC core diameter used for calculating coating properties.

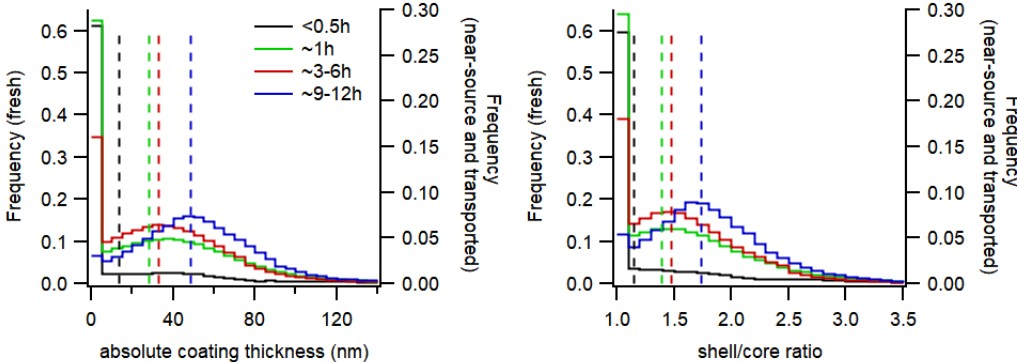

**Figure 4:** Distributions of measured coating thickness of BC-containing particles in sampled smoke plumes at different ages, in terms of shell/core ratios (left) and absolute coating thickness (right). The first bin ($D_P / D_C = 1$, uncoated particle) contains particles with measured scattering less than that predicted for an uncoated core (equivalent to $D_P < D_C$).





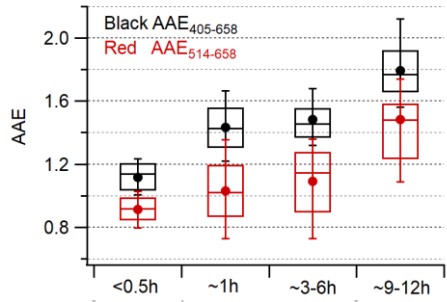

**Figure 5: The measured AAE$_{405-658}$ (black) and AAE$_{514-658}$ (red) in sampled smoke plumes at different ages. The box-and-whisker plots represent the 10th percentile, 25th percentile, median, 75th percentile and 90th percentile, the dot markers represent mean values.**

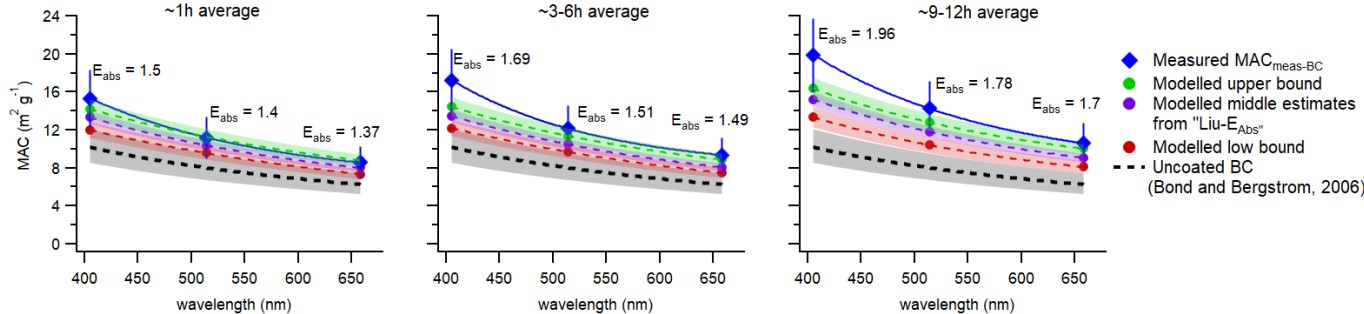

**Figure 6: The measured and modelled MAC values at 405, 514 and 658 nm in sampled smoke with different ages, including the measured MAC$_{meas-BC}$ (blue markers), the low (red circle markers) and upper (green circle markers) limits and approximately middle estimates (purple circle markers, from "Liu-E$_{Abs}$") of modelled MAC (MAC$_{modelled}$). The uncertainties of MAC$_{meas-BC}$ include the instruments uncertainties and the fit errors. The uncertainties of MAC$_{modelled}$ are from the Monte Carlo analysis as in Taylor et al. (2020). The black dashed lines and shaded areas represent the MAC and uncertainties of uncoated BC reported by Bond and Bergstrom (2006).**

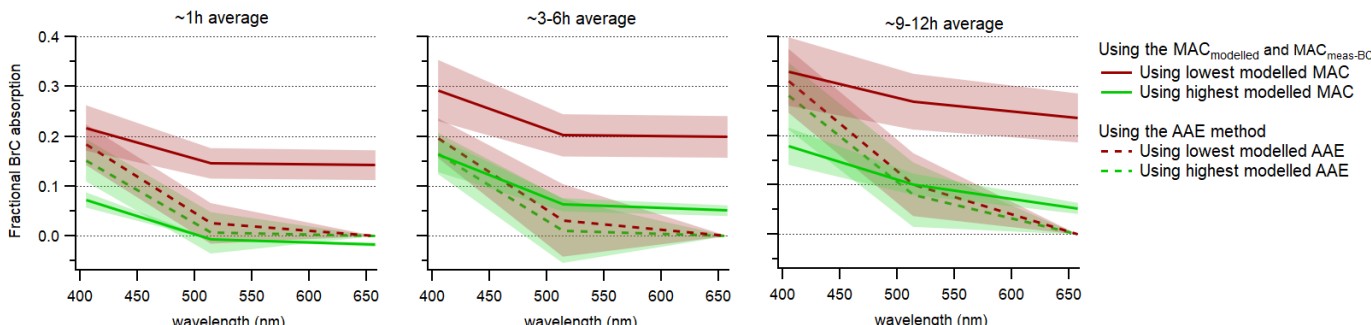

**Figure 7: The simulated fractional contribution of BrC to total aerosol absorption, including different wavelengths and sampled smoke with different ages. The solid red and green lines represent the mean results calculated from the MAC$_{modelled}$ and measured MAC$_{meas-BC}$, using the lowest and highest modelled MAC respectively. The dashed red and green lines represent the mean results calculated from the AAE methods, using the lowest and highest modelled AAEs. The shades represent the uncertainties.**



**Table 1. The fundamental information and modified combustion efficiency (MCE) of sampled smoke plumes.**

| Sample age | < 0.5 h | ~ 1 h | ~ 3–6 h | ~ 9–12 h |
|---|---|---|---|---|
| Date | 01/03/2017 | 01/03/2017 | 01/03/2017 | 02/03/2017 |
| Flight number | C005 | C005 | C006 | C007 |
| Aircraft Altitude (AGL, m) | 380 – 1486 | 745 – 1980 | 1642 – 1728 | 1482 – 1780 |
| Ambient Temperature (°C) | 29.3 ± 3 | 24.5 ± 3 | 23.1 ± 0.3 | 22.5 ± 0.5 |
| Ambient Relative humidity (%) | 16 ± 2 | 18 ± 2 | 19 ± 1 | 25 ± 3 |
| MCE | 0.94 – 0.96 | 0.94 ± 0.01 | 0.94 ± 0.01 | 0.94 ± 0.08 |

**Table 2. The ERs of BC and OA and some chemical information in sampled smoke.**

| | < 0.5 h | ~ 1 h | ~ 3–6 h | ~ 9–12 h |
|---|---|---|---|---|
| $\Delta BC/\Delta CO$ ($\mu g\ cm^{-3}/\mu g\ cm^{-3}$) | 0.016 ± 0.003 (min – max: 0.012 – 0.021) | 0.018 ± 0.004 | 0.017 ± 0.003 | 0.013 ± 0.003 |
| $\Delta OA/\Delta CO$ ($\mu g\ cm^{-3}/\mu g\ cm^{-3}$) | 0.071 ± 0.032 (min – max: 0.045 – 0.101) | 0.079 ± 0.030 | - | 0.066 ± 0.027 |
| OM/OC | 1.52 ± 0.03 | 1.68 ± 0.05 | - | 2.11 ± 0.04 |
| $\Delta OA/\Delta BC$ | 7.2 ± 0.9 | 5.6 ± 0.5 | - | 5.9 ± 0.4 |
| $\Delta OC/\Delta BC$ | 5.0 ± 0.6 | 3.5 ± 0.3 | - | 2.9 ± 0.2 |

Note: OA information was lost in the transported smoke at an age of ~ 3–6 h, as there was no AMS data for the period.