# Peer review of "Rapid transformation of ambient absorbing aerosols from West African biomass burning"

_Atmospheric Chemistry and Physics, 2021_

## Author Response (AR1)

**Response to reviewers**

**Firstly, we would like to thank the referees for their important comments, which we have addressed below. The original comments are in black, our replies are in blue and the changes in original manuscript are in red.**

**Anonymous Referee #1**

Wu et al. presented the field aircraft campaign results in investigating a half-day evolution of flaming burning dominated smoke aerosols over the Senegal region. The chemical and optical properties of the smoke aerosols during transport were monitored and analyzed to depict the rapid transformation of the absorbing particles, and they found increasing contribution from secondary BrC in bulk aerosol absorption during the initial aging procedure. There is enormous amount of publications in studying the emission and evolution of biomass burning related light absorbing carbonaceous particle. This study is surely a good addition. I suggest publication after addressing the following minor comments.

**Minor comments:**

**1.** More background information is suggested to provide in the manuscript, including aging environment, exact time profile for the smoke transport and flights (morning or afternoon). Figure S2 should be better moved to manuscript.

We have added the recommended information (ageing environment, time range) in the manuscript. We have moved Figure S2 from the supplementary to the manuscript.

Overall, these selected smoke plumes had a discrete range in plume age, from about <0.5 to ~12 h, which provided an opportunity to study the evolution of BB aerosol properties during the first ~12 h of transport. For the near-source (C005) and transported smoke with an age of ~3–6 h (C006), the smoke had undergone only daytime (photochemical) ageing after emission. For the transported smoke with an age of ~9–12 h (C007), the smoke had undergone ~0–2 h of night-time ageing, followed by further daytime ageing.

**Table 1. The fundamental information and modified combustion efficiency (MCE) of sampled smoke plumes.**

| Sample age | < 0.5 h | ~ 1 h | ~ 3–6 h | ~ 9–12 h |
|---|---|---|---|---|
| Flight number | C005 | C005 | C006 | C007 |
| Date | 01/03/2017 | 01/03/2017 | 01/03/2017 | 02/03/2017 |
| Sampling range (UTC) | 12:37:36 to 13:27:30 | 13:09:34 to 13:37:10 | 17:47:00 to 17:55:00 | 15:57:12 to 16:07:32 |
| Aircraft Altitude (AGL, m) | 380 – 1486 | 745 – 1980 | 1642 – 1728 | 1482 – 1780 |
| Ambient Temperature (°C) | 29.3 ± 3 | 24.5 ± 3 | 23.1 ± 0.3 | 22.5 ± 0.5 |
| Ambient Relative humidity (%) | 16 ± 2 | 18 ± 2 | 19 ± 1 | 25 ± 3 |
| Estimated Source Burn Area | ≤1 km$^2$ | ≤1 km$^2$ | ≤1 km$^2$ | ≤1 km$^2$ |
| Estimated Source Burn time (UTC) | 11:25 to 13:17 | 11:25 to 13:17 | 11:51 to 13:17 | 03:09 to 07:32 |
| MCE | 0.94 – 0.96 | 0.94 ± 0.01 | 0.94 ± 0.01 | 0.94 ± 0.08 |

**2.** In discussing organic characters as measured by AMS systems, it should be described how to check the possible influence of so-called Lieber effects/artefacts (i.e., Pieber et al., ES&T, 2016; Freney et al., AST, 2019), especially for inorganic salts contributing to a considerable portion of the bulk aerosol.

Thanks for the suggestion.

Previous studies show that $NH_4NO_3$ and other nitrate salts (i.e., $KNO_3$ and $NaNO_3$), as well as $(NH_4)_2SO_4$, can lead to the formation of non-OA $CO_2^+$ (i.e., $N_2O^+$) in the Aerodyne AMS. This observed interference will introduce biases in OA mass and chemical composition measurements, particularly regarding the aerosol oxygen content ($f44$, O/C) (i.e., Pieber et al., 2016; Freney et al., 2019). From our AMS calibrations using the $NH_4NO_3$ and $(NH_4)_2SO_4$, we performed an orthogonal distance regression linear fit between $CO_2^+$ and $NO_3$ signals and $CO_2^+$ and $SO_4$ signals. The slopes were determined to be 1.8% for $NH_4NO_3$ ($CO_2^+$ and $NO_3$ signals) and 0.5% for $(NH_4)_2SO_4$ ($CO_2^+$ and $SO_4$ signals). The $NO_3$/OA ratio was (3–7) % and the $SO_4$/OA ratio was (0.4–9) % during the MOYA campaign. The biases of $NH_4NO_3$ and $(NH_4)_2SO_4$ for OA mass, $f44$ and O/C are expected to be less than 1%, due to the small slopes between $CO_2^+$ and $NO_3$ signals for $NH_4NO_3$ and $CO_2^+$ and $SO_4$ signals for $(NH_4)_2SO_4$, and the limited inorganic fractions. Thus, the minor interference of non-OA $CO_2^+$ from the inorganic species are not considered in OA mass and OA composition calculations.

**3.** Did the authors consider the influence of dynamic inorganic mixing in absorption characterization of smoke aerosol?

During MOYA, we observed enhanced $MAC_{meas-BC}$ (the total aerosol absorption normalized to BC mass) upon half-day ageing, due to the lensing effect of increasingly thick coatings on BC and the absorption of BrC. In Sect. 3.3, we explained that "the inorganic species (nitrate and sulfate), formed from the oxidation of emitted gaseous $NO_x$ and $SO_2$ after emission, would also condense onto existing particles". This inorganic mixing would partly contribute to the lensing effect of coatings on BC. However, it is beyond the scope of this work to separate and quantify this lensing effect of inorganic species as BC coatings.

**Specific comments:**

1 – **Page 2 Line 60** – change to "though both estimates are associated with considerable uncertainties." Accepted

2 – **Page 2 Lin 61** – delete "than this". Accepted

3 – **Page 3 Line 75**– add "coated" or "internal mixed" before "BrC". Accepted

4 – **Page 5 Line 161**– where was the impactor installed? Ahead of the PAS?

The impactor was installed ahead of the PAS, we have added it into the manuscript.

An impactor upstream of the PAS removed particles with aerodynamic diameters > 1.3 μm.

5 – **Page 6 Line 164** – check equation 1, AAE is positive value.

We have amended the equation, the equation is right, AAE is positive value.

$$AAE = -\frac{\ln(B_{Abs}(\lambda_2)) - \ln(B_{Abs}(\lambda_1))}{\ln(\lambda_2) - \ln(\lambda_1)} \tag{1}$$

**6 – Page 7 Line 198 –** "Further details in the processing ……" Accepted.

We have rephrased this statement to "Further details concerning the processing…"

**7 – Page 8 Line 248 –** MCE of 0.9 is a simple threshold to classify burning phase, MCE of 0.9 and beyond roughly indicates flaming burning dominated in a fire event.

We have rephrased the sentence.

An MCE > 0.9 is commonly used to indicate BB smoke predominantly influenced by combustion during the flaming phase, whereas MCE < 0.9 indicates that the BB smoke is primarily emitted from smouldering phase combustion.

**8 – Page 10 Line 307 –** chemical formulas for these specific ions should be added.

We have added the chemical formulas for these specific ions.

The ion peak at m/z 60 $(C_2H_4O_2^+)$ is attributed to levoglucosan-like species, which has been accepted as a marker of BB pyrolysis products (Schneider et al., 2006). The m/z 43 $(C_3H_7^+)$ and 57 $(C_4H_9^+)$ markers are from the fragments of saturated hydrocarbon compounds and long alkyl chains and are good indicators of fresh aerosols (Alfarra et al., 2007). The m/z 43 marker can also come from oxidized functionalities such as aldehydes and ketones (Alfarra et al., 2007). The m/z 44 is the signal of $CO_2^+$ ion from carboxylic acid groups and organo-peroxides and suggests the presence of oxygenated organic compounds (Aiken et al., 2008).

**9 – Page 18 Line 560-561 –** Work by Li et al. (2020) was nighttime $NO_3$ radical reaction that enhanced light absorption by BB-BrC, the reaction pathway should be different from the photochemical aging in the manuscript. Saleh et al. (2013) reported secondary BrC formation in photochemical aging of BBOA, but these secondary BrC had less absorption than primary BrC at wavelength beyond 400 nm. Commonly, OH radical photochemical oxidation diminishes light absorption by primary BrC, unless NOx involving to prohibit the bleaching via new chromophore formation (Li et al., 2019. DOI:10.5194/accp-18-1-2018).

Thanks for the suggestion, we have rephrased this part.

Laboratory studies have provided evidence that secondary OA formed by photo-oxidation of BB emissions can contain BrC. These aerosols containing secondary BrC exhibit a stronger wavelength dependent absorption than primary BrC, absorbing light less efficiently at long visible wavelengths and being more absorptive at short visible and near-UV wavelengths (Saleh et al., 2013). Li et al. (2019) also reported that simulations under high-NOx conditions can enhance the formation of BrC and light absorption for tar ball aerosols from BB, relative to OH photooxidation in the absence of NOx. These observations suggest that secondary BrC formation could counteract photobleaching to eventually re-establish absorption enhancement of BrC. For the transported smoke sampled at ~9–12 h, the BB aerosol is likely to have undergone ~0–2 h of night-time ageing first and then daytime ageing. The night-time chemistry involving the $NO_3$ radical reaction with BB OA can also increase the BrC absorption efficiency over the UV-vis range (Li et al., 2020).

**10 – Page 20 Line 634-635 –** confused. Do you mean that 20% of the observed aerosol is background one after half-day transport?

After emission, BB plumes will mix with nearby background air and will be diluted. Nearby background air out of the plume consisted of regional haze and aged BB emissions. For the smoke plume at source (<0.5 h), ~1 h and ~3–6 h, the effect of mixing with background air on plume aerosols is expected to be negligible, as the aerosol loadings in plumes were tens to hundreds of times greater than those in the nearby background air. Aerosol concentrations and light absorption coefficients for smoke at ~9–12 h were elevated only by a factor of ~5 compared to nearby background levels. Based on the method of Murphy et al. (2009), ~20 % of the observed aerosol at ~9–12 h is likely due to the mixing with background aerosol. Therefore, yes, we do mean that background air may contribute to ~20% of the observed aerosol in the plume. As the aerosol properties in nearby background air (Table S2) were similar to the smoke aerosols at ~9–12 h, this mixing still would not affect smoke aerosol properties significantly. Above all, we want to clarify that the evolution of BB aerosol properties reported in this study is dominated by chemical and physical processing during transport, excluding the background effects.
We have rephrased the previous line 634-635, to make this clearer.
Aerosol concentrations and light absorption coefficients in smoke sampled at ~9–12 h were both elevated by a factor of ~5 compared to nearby background levels. Based on the method of Murphy et al. (2009), background air contributed to ~20% of the observed aerosol in the plume due to the mixing.

**11 – Page 21 Line 661 –** levoglucosan is not chromophore, the positive correlation between absorption and marker fragment ratio indicated primary BrC emission from biomass burning, and the aging played a major bleaching role.
Thanks for the suggestion, we have rephrased this part.
Lack et al. (2013) sampled near-source smoke emitted from a large Ponderosa Pine forest fire near Boulder, Colorado. They found that the $AAE_{404-658}$ and non-BC absorption at 404 nm were positively correlated to the $f60/f44$ ratio, indicating that their measured BrC was linked to primary OA and photobleaching (through photolysis and photo-oxidation) played a major role during ageing.

**12 – Page 21 Line 665 –** confused. "Chemical reaction loss" means absorption decrease due to reaction or levoglucosan decomposition indicated by f60 decrement in aging?
The chemical reaction loss in the manuscript refers to the photobleaching (photolysis and photo-oxidation) of BrC, which results in the loss of BrC and therefore a reduction in BrC absorption. We have amended this sentence to emphasise that we are referring to photolysis and photo-oxidation processes, which now reads:
This case study also indicates a major contribution of primary OA to BrC and suggested that photobleaching loss (through photolysis and photo-oxidation) dominated BrC evolution.

**13 – Page 21 Line 670 –** do you mean smoldering burning is more efficient in primary BrC emission?
Yes, we agree that smouldering burning is more efficient in primary BrC emissions than flaming burning, as the smouldering-phase burning is well known to favour the formation of OA rather than BC. Laboratory simulations by McClure et al. (2020)

suggest that BB emissions with primary compositions dominated by organic matter are more likely to contain more significant BrC than those dominated by BC content. We have added the following amendments to the manuscript:

The case study by Forrister et al. (2015) investigated smouldering-controlled burning at source, which yielded much higher initial OA/BC ratios (> 100) than the flaming-controlled burning in this study (~7) and gave larger initial AAE. Smouldering burning is generally more efficient in primary BrC emissions than flaming burning, as the smouldering-phase favours the formation of OA rather than BC, and BB emissions with primary compositions dominated by organic matter are more likely to contain higher fractional concentrations of BrC than those dominated by BC (e.g., McClure et al., 2020).

**Anonymous Referee #2**

This manuscript by Wu et al. presents aircraft measurements of ageing smoke plumes of agricultural and savannah flaming fires in the Senegal region. The measurements characterized the evolution of size distributions, chemical composition, and light-absorption properties of the aerosol emissions for plume ages up to 12 hours. The major findings include (1) observed significant chemical transformation of the organic aerosol (OA) but without increase in OA loading, which is attributed to a combination of primary OA oxidation, secondary OA formation, and primary OA evaporation due to dilution; and (2) increase in brown carbon absorption with atmospheric age. The paper is well-written and is a valuable contribution to the atmospheric chemistry literature. I have just one major comment on the optical calculations, detailed below.

**Major comments:**

The use of different models to calculate MAC values and derive BrC contribution to absorption does not seem to add useful insight to the analysis and conclusions regarding the evolution of BrC absorption in the plumes. With absence of detailed information on particle morphology and actual MAC_BC, there is a lot of uncertainty that goes into these MAC calculations. (1) The calculations are based on the assumption that MAC_BC = 7.5 $m^2$/g at 550 nm applies to the measurements in this study. This alone can lead to substantial uncertainty. Any over/underestimation in BC mass concentration measurements and/or over/underestimation in light-absorption measurements would lead to misattribution of absorption enhancement to lensing and/or BrC absorption. (2) It is not clear that the experimental conditions on which the empirical models (Liu, Wu, Chak) were based apply to the aerosol in this study.

Thanks for the comment.

(1) We agree that the detailed information on particle morphology and actual MAC of uncoated BC is absent. In our optical simulations, we introduced an alternative metric of the mass ratio of non-BC to BC (MR= $M_{non-BC}/M_{BC}$) (Liu et al., 2017), and we generated a 2-D mixing state distribution of MR versus $M_{BC}$ (Taylor et al., 2020). So, the simulations in this study are framed in terms of MR versus $M_{BC}$. The advantage of using the MR as a metric here to calculate the MAC of coated BC is that *it does not assume anything about particle morphology*. The disadvantage is that more explicit optical models than the models used in this study cannot apply the MR. For some parameterisations which are based on empirical fits to the bulk absorption enhancement ($E_{Abs}$) for BC particles of different mixing states (Liu et al., 2017; Chakrabarty and Heinson, 2018; Wu et al., 2018), we multiplied the modelled "$E_{Abs}$" by the MAC of uncoated BC ($MAC_{BC}$) from Bond

and Bergstrom (2006). We agree that this would result in some uncertainty based on the assumption that is summarised from previous literatures ($MAC_{BC}$ = 7.5 m$^2$ g$^{-1}$ at $\lambda$ of 550 nm, with AAE = 1). However, these empirical models have been shown to produce MAC and AAE values of BC-containing particles in relatively good agreement with measurements in previous studies (e.g., Liu et al., 2017; Taylor et al., 2020). Thus, we selected these optical models in this study.

We agree that any over/underestimation in BC mass concentration and light-absorption measurements would lead to misattribution of absorption enhancement to lensing and/or BrC absorption. So, we have assessed the uncertainties in modelled values from these different optical models using a Monte Carlo analysis (Taylor et al., 2020), which considered the uncertainties from different input variables, including BC mass, MR, non-refractory material concentrations and other variables. We also have considered the BC mass and PAS (absorption) measurement uncertainty in BrC attribution. The uncertainties in our model calculations (from the Monte Carlo analysis), measurements and estimated BrC attribution are included in related plots (Figure 7 and 8 in updated manuscript)).

(2) The empirical models (Liu, Wu, Chak) used in this study have been explained explicitly in the supplementary material. We have also added the brief information in Sect. 2.3 in the manuscript.

Briefly, Liu et al. (2017) conducted ambient measurements of $E_{Abs}$ for BC-containing particles from different combustion sources and also a laboratory chamber study of fresh and aged diesel soot. They made an empirical correction of $E_{Abs}$ to the core/shell Mie models based on laboratory and atmospheric observations. Wu et al. (2018) also introduced an empirical correction of $E_{Abs}$ to core/shell Mie models based on both ambient measurements and Aggregate model results which were constrained by BC micromorphology. Chakrabarty and Heinson (2018) integrated modelled results and observational findings to establish scaling relationships for $E_{Abs}$ and $MAC_{BC}$ as a function of BC mass and coating thickness.

The message on the evolution of BrC absorption with plume age, which I believe is an interesting one, can be delivered more cleanly by just relying on MAC_measured_BC and AAE. Instead of Figure 6 (which is a bit hard to follow), I would add another panel to Figure 5 that shows box plots of MAC_measured_BC at different ages.

Thanks for the suggestion, we have added the figure to show the MAC_measured_BC at different ages.

[Figure]

**Figure 6: (a) The measured AAE$_{405-658}$ (black) and AAE$_{514-658}$ (red) in sampled smoke plumes at different ages. The box-and-whisker plots represent the 10th percentile, 25th percentile, median, 75th percentile and 90th percentile, the dot markers represent mean values. (b) The measured MAC values (markers) with uncertainties (shades) at 405, 514 and 658 nm in sampled smoke with different ages, the black dashed lines and shaded areas represent the MAC and uncertainties of uncoated BC reported by Bond and Bergstrom (2006).**

As for BrC contribution, I believe that the simple AAE attribution method (with absence of detailed information to allow more involved modelling) is the best that could be done. In fact, the AAE method seems to yield more reasonable results (in terms of wavelength-dependence of fractional BrC absorption) than the modelling methods which show very weak wavelength-dependence of fractional BrC absorption.

The upper bounds of BrC contribution at 658 nm calculated using the lowest $MAC_{modelled}$ and $MAC_{meas-BC}$, were ~ 15 – 25%. The low bounds of the estimated contribution fraction at 658 nm calculated using the highest $MAC_{modelled}$ and $MAC_{meas-BC}$ were minor (<5%) throughout the transport time. Previous studies have observed that BrC absorption decreases significantly from near-UV to visible ranges and is negligible close to the wavelengths of 700 nm (e.g., Laskin et al., 2010; Liu et al., 2015). So, we have clarified in the manuscript that the upper bound estimates from the method of comparing $MAC_{modelled}$ and $MAC_{meas-BC}$ are likely to be overestimated, and the low bounds are more reliable. In addition, the low bounds of BrC attribution at 405 nm and 514 nm from the $MAC_{modelled}$ vs. $MAC_{meas-BC}$ are mostly within the uncertainty in calculated BrC fraction from the AAE method. We think that it is reasonable to combine the ranges from AAE methods and the low bounds from the $MAC_{modelled}$ vs. $MAC_{meas-BC}$. We are not convinced that the modelling approaches are worse. Indeed, there are also drawbacks to the analyses that rely solely on the AAE attribution approach and we are strong of the view that the combined results of the AAE attribution method and the empirical modelling methods (as validated in previous studies, albeit for different source regions of BBA) provide an improved foundation on which to draw conclusions concerning BrC attribution.

**Specific comments:**

**1 – Line 169 –** the statement about inverting the SMPS data is not clear.

The aerosol group in Lund has developed the inversion algorithms for the SMPS data (Zhou, 2001). These programs are written in the LabVIEW graphical programming language, since the SMPS data acquisition systems are operated in the LabVIEW interface. By taking into account the sampling line losses, bipolar charging probabilities, calibrated DMA transfer functions regarding DMA diffusion broadening and losses and CPC counting efficiencies, the inversion program calculates the theoretical kernel transfer matrix and inverts mobility concentrations to an aerosol size distribution.

The SMPS data were inverted using the inversion algorithms developed by Zhou (2001). The inversion program inverts mobility concentrations to an aerosol size distribution (dN/dlogDp vs. Dp). The analysed SMPS data were based on a ~ 1 min averaging time only during straight and level runs when AMS and SP2 concentrations generally varied less than 30 %.

**2 – Line 224 –** It is not clear why modeled MAC instead of B_Abs was used to calculate AAE.

MAC is the output from our different models. The modelled MACs (at 405, 514 and 658 nm) were determined by the ratio of the modelled absorption cross-section ($B_{Abs}$) to the total BC mass. The AAE between two wavelengths can be determined by Eqn. (1). By substituting in the definition of MAC (as the ratio of $B_{Abs}$ to the BC mass) into Eqn. 2, the mass terms cancel, and Eqn. 1 is derived. The Eqn. 1 and Eqn. 2 give the same results. Here, we use the modelled MAC in Eqn. 2 instead of the $B_{Abs}$ in Eqn. (1).

$$AAE = - \frac{\ln(B_{Abs}(\lambda_2)) - \ln(B_{Abs}(\lambda_1))}{\ln(\lambda_2) - \ln(\lambda_1)} \tag{1}$$

$$AAE = - \frac{\ln(MAC(\lambda_2)) - \ln(MAC(\lambda_1))}{\ln(\lambda_2) - \ln(\lambda_1)} \tag{2}$$

**3 – Line 233 –** replace "some" with a number (more quantitative).

We have rephrased this sentence.

On 1 March 2017, the ARA (flight C005) flew over a selected MODIS-detected fires repeatedly (Fig. 1b) and sampled fresh plumes at different heights ($\sim$400 – 1500 m) during the plume rise stage.

**4 – Line 234 –** what is the assumption that the plumes are less than 0.5 hours old based on?

We sampled the fresh plumes by positioning the ARA directly over the active fires. We analysed the densest plume transect which is denoted as source emissions. Based on the distance from the fires and the average wind speed measured by the aircraft, the reaction time between emission and the measured densest transect was less than < 0.5 h.

The fresh plumes were sampled by positioning the ARA directly over the active fires. We analysed the densest plume transects which are denoted as source emissions, and the fresh plumes were assumed to be less than 0.5 h old.

**5 – Line 321 –** add "aerosol" after "secondary organic". Accepted

**Anonymous Referee #3**

This paper describes aircraft measurements from three flights in west Africa that sampled biomass burning. The authors examine the aerosol optical properties as a function of transport age over 0 - 12 hours. The paper is well-written and well-organized.

**Major comments:**

**1.** - Section 2: It would be useful to provide a basic overview of the campaign and the fires sampled in a few sentences. Specifically: What were the dates of the study? How many total flights were made? What was the aircraft duration? What were the criteria for selecting these three flights for this study?

Thanks for the suggestion, we have added the recommended information.

The research flights during MOYA-2017 (Methane Observation Yearly Assessment-2017) were made by the UK Facility for Airborne Atmospheric Measurements (FAAM), using the BAe-146 Atmospheric Research Aircraft (ARA). (Flight number, date and duration) *A total of six flights (designated flight labels from C003 to C008) took place between 27 February and 3 March 2017, with the precise timings and objectives of these flights provided in Table S1.* The aircraft was equipped with a range of in situ instruments to measure aerosol composition, size distribution and optical properties, as well as trace gas concentrations and standard meteorological variables. A further description of the MOYA-2017 campaign is reported by Barker et al., 2020. Tracks of the flights (with flight numbers labelled from C005 to C007) used in this study are shown in Fig.1a. (Selection criteria) *These selected flights focused on freshly emitted plumes from wildfires over the Senegal area, in*

*addition to aged smoke transported southwest over the continent and the Atlantic Ocean.* Nearby background air out of the plume was also sampled. Detailed information about the selected smoke plumes is provided in Sect. 3.1. The main instruments used in this study are described below.

**Table S1. The fundamental information of the MOYA aircraft flights.**

| Flight | Date | Time | Duration | Objectives |
|--------|------|------|----------|------------|
| C003 | 27/02/2017 | 09:15:50 to 13:55:40
15:35:11 to 19:15:37 | 4:39:50
3:40:26 | Transit flight |
| C004 | 28/02/2017 | 11:51:13 to 15:36:15 | 3:45:02 | Sampling fresh plume
(optical instrument issues) |
| C005 | 01/03/2017 | 10:57:57 to 14:56:53 | 3:58:56 | Sampling fresh plume |
| C006 | 01/03/2017 | 16:32:13 to 20:10:33 | 3:38:20 | Sampling transported plume |
| C007 | 02/03/2017 | 12:48:55 to 16:54:58 | 4:06:03 | Sampling transported plume |
| C008 | 03/03/2017 | 07:35:36 to 10:38:03
12:20:16 to 15:46:01 | 3:02:27
3:25:45 | Transit flight |

**2.** - Section 2.2: What is the minimum detectable fire size for MODIS? Were most of the fires in the region detected?

Thanks for the suggestion, we have added the recommended information.

MODIS routinely detects both flaming and smouldering fires with a minimum size of 1000 $m^2$. Under very good observation conditions (e.g., with the satellite in a near-nadir viewing geometry with respect to the fire, relatively homogeneous land surface), one tenth of this size can be detected for flaming fires. Most of the fires in this region could be detected by the MODIS (Giglio et al., 2016).

**3.** - Section 3.1: What was the fuel for the agricultural fires? What was the burn area? How long did the fires persist? How similar were the fuels and burn conditions for the different fires? These are important questions because the analysis of different smoke ages represents different fires sampled during different flights. If the fire conditions differed between the flights, that will affect the trends.

The main crop in this area is millet (Fare et al., 2017). We have added the recommended information in Sect. 3.1.

The fire areas are mainly a mixture of agricultural stubble (mostly millet crop) and wooded savannah (deciduous forest matter and savannah grasses) (Roberts et al., 2009; Fare et al., 2017; Barker et al., 2020).

We estimated the burn area and persistence time of source fires for sampled smoke plumes from the MODIS. The burn areas of source fires were less than a pixel size (1 $km^2$) in MODIS. The burn times of source fires were estimated from the start and end time that we can detect the fires from the MODIS. This information has been added to Table 1.

We have clarified in Sect. 3.1, the calculated MCEs (Table 1) of selected smoke plumes at different transport ages were in a small range of 0.94 to 0.96. The backward analysis also indicates that they originated from nearby fire areas that have the same fuel type, which is mainly a mixture of agricultural stubble (mostly millet crop) and wooded savannah (Roberts et al., 2009; Fare et al., 2017; Barker et al., 2020). Both the similar fuel and MCE suggest that the smoke plumes we selected from different flights are comparable in terms of the initial aerosol properties at source.

**4.** - Figure 1 shows that transects for each flight were all sampled at the same distance downwind. Why not make multiple downwind transects at increasing distance from the source?

It's a pity that we didn't sample more plume transects at different distances downwind. We will consider this in future studies.

**5.** - Section 3.3: What was the uncertainty of the SMPS scans? Due to the slow time response, it is more typical to use an optical particle counter for aircraft measurements. Was there a reason that the SMPS was used?

The uncertainty of the SMPS scans is ~33%. We have added this information in Sect. 2.1.

We did not have the optical particle counter on BAe-146 ARA during the MOYA campaign. The PCASP (Passive cavity aerosol spectrometer probe, 0.1–3μm) and GRIMM (GRIMM sky optical particle counter, 0.25–32μm) are usually used to provide accumulation mode measurements on the FAAM aircraft, but were not fitted and could not cover the size range of interest in this work. The only instrument onboard for aerosol size distribution measurements, focusing on the smaller diameter size range (20–350nm), is the SMPS we analysed.

**Minor comments:**

**1. - Line 35-37:** The aerosols aren't evolving in the fires; they are evolving downwind. This sentence might be clearer as "Different treatments of absorbing aerosol properties from smoldering and flaming combustion and their downwind evolution should be considered..."

Accepted

Different treatments of absorbing aerosol properties from different types of fires and their downwind evolution should be considered when modelling regional radiative forcing.

**2. - Lines 56-57:** Consider including earlier references.

Thanks for the suggestion, we have added earlier references.

However, certain types of OA, known as "brown carbon" (BrC) also absorb solar radiation in the near-ultraviolet (near-UV, 300−400 nm) and visible (400−700 nm) ranges, although this absorption is strongly wavelength dependent compared to the absorption spectrum for BC (*Bond and Bergstrom, 2006; Ramanathan et al., 2007;* Laskin et al., 2015).

**3. - Line 64:** "The initial relative contribution of OA and BC varies...." It is unclear if you mean the mass contribution or the absorption contribution.

Accepted

The initial relative *mass* contribution of OA and BC varies widely with fuel type and combustion conditions, as does the corresponding initial aerosol size distribution (Vakkari et al., 2014).

**4. - Line 75:** Consider including older Lack and Langridge references?

We accepted the suggestion of older references, but use another two works by Lack et al., 2012a, b.

The MAC of coated BC may be enhanced by a lensing effect induced by the coatings and/or the absorption from internally mixed BrC (*Lack et al., 2012a, b*; Healy et al., 2015).

**5. - Line 97:** Could the acronym "MR" be eliminated and replaced with $M_{nonBC}/M_{BC}$? By the time it appeared here, I had to search for the definition again.

We could not find the "MR" in line 97 that the reviewer refers to.

The first time "MR" appeared previously is in line 197 (Sect. 2.3). The "MR" has been used as an academic abbreviation for the mass ratio of non-BC to BC in many previous optical modelling studies (e.g., Liu et al., 2017; Chakrabarty and Heinson, 2018; Wu et al., 2018; Taylor et al., 2020). We prefer to keep the "MR" in this study. To help readers, we have repeated the definition of "MR" in the manuscript.

**6. - Lines 396-397:** What RI is assumed for BC?

Simulations by Liu et al. (2018) employed a RI of $1.8 + 0.6i$ for BC.

Simulations by Zhang et al. (2020) employed a RI of $1.85 + 0.71i$ for BC.

We have added this information in the manuscript.

**7. - Figure 1:** Color the MODIS-detected fires according to the three flights (blue, green, pink).

**8. - Figure 1** Caption: Change "1-day back trajectory of selected sampled smoke over the Atlantic Ocean during flight C006 (c) and C007 (d)" to "1-day back trajectory of sampled smoke from flight C006 (c) and C007 (d)" because it sounds like the back trajectory is over the Atlantic Ocean but its actually the flight that was over the Atlantic Ocean.

We have removed the MODIS fire in Fig. 1a, to only show the flight tracks. For each flight (Fig. 1b for C005, Fig. 1c for C006 and Fig. 1d for C007), the corresponding MODIS fire is marked separately.

[Figure]

**Figure 1: (a) Tracks of the flights (labelled from C005 to C007) used in this study. (b) The selected fresh and near-source plumes, with the spatial distribution of MODIS-detected fires during flight C005. (c-d) 1-day back trajectory of sampled smoke selected from flight C006 (c) and C007 (d), marked (black crosses) with every 3h increment. The MODIS-detected fires are also shown in the plots (as observed 3 - 12 h before the sampling period).**

**9. - Figure 2:** Upper whiskers are hidden on the bar chart.

Adding the upper whiskers (see Fig. 3a below) makes the plot harder to read, especially for the inorganic components. And, the upper and bottom whiskers are symmetric. We prefer to only put the bottom whiskers in the bar chart. We prefer to keep the original plot.

[Figure]

**Figure 3: (a) The average chemical compositions of sampled smoke submicron aerosols at different ages (left axis), and the black solid circles represent the average O:C ratios of OA in sampled smoke (right axis). The whiskers represent one standard deviation. (b) The fractional signals f44 vs. f60 and f44 vs. f43 of sampled smoke aerosols in our study. The dashed vertical line represents the background of f60 (0.3%) in environments not influenced by BB, as recommended by Cubison et al. (2011). The dashed-dot line passing through measurement data indicates the general trend in f44 vs. f60 with aerosol age.**

**10. - Figure 4:** It is unclear which traces are assigned to the left and right axes.

Thanks for the suggestion. We have made corrections to the left and right axes labels, to make them clearer.

[revised manuscript text omitted]